# DreamSwapV: Mask-guided Subject Swapping for Any Customized Video Editing

**Weitao Wang**[1,2] **Zichen Wang**[2] **Hongdeng Shen**[2] **Yulei Lu**[2] **Xirui Fan**[2]
**Suhui Wu**[2] **Jun Zhang**[2*] **Haoqian Wang**[1*] **Hao Zhang**[2†]
[1]Tsinghua University  [2]ByteDance
wangwt23@mails.tsinghua.edu.cn

## Abstract

With the rapid progress of video generation, demand for customized video editing is surging, where subject swapping constitutes a key component yet remains under-explored. Prevailing swapping approaches either specialize in narrow domains—such as human-body animation or hand-object interaction—or rely on some indirect editing paradigm or ambiguous text prompts that compromise final fidelity. In this paper, we propose DreamSwapV, a mask-guided, subject-agnostic, end-to-end framework that swaps any subject in any video for customization with a user-specified mask and reference image. To inject fine-grained guidance, we introduce multiple conditions and a dedicated condition fusion module that integrates them efficiently. In addition, an adaptive mask strategy is designed to accommodate subjects of varying scales and attributes, further improving interactions between the swapped subject and its surrounding context. Through our elaborate two-phase dataset construction and training scheme, our DreamSwapV outperforms existing methods, as validated by comprehensive experiments on VBench indicators and our first introduced DreamSwapV-Benchmark.

## 1 Introduction

The recent development of video generation technologies is enabling a broader audience to engage in customized content creation, also stimulating demand for editing real or generated videos. Users increasingly desire to introduce specific subjects into videos following the original motion trajectory, necessitating the video subject swapping task. Given a source subject in the video and a target subject to be inserted, the ideal outcome is that the target is seamlessly integrated into the original video, preserving its own appearance details, following the source motion, and interacting naturally with the surrounding context.

However, existing video editing methods exhibit critical limitations: On one hand, domain-specific approaches are confined to particular subjects: animation techniques like MagicAnimate (Xu et al., 2024b) and Animate Anyone 2 (Hu et al., 2025a) only support swapping human characters; Human-Object Interaction (HOI) methods like AnchorCrafter (Xu et al., 2024c) and DreamActor-H1 (Wang et al., 2025) are limited to specific scenarios (e.g. livestreaming, E-commerce) for swapping hand-held objects; thereby restricting their broader applicability. On the other hand, general-purpose video editing methods also face several challenges: (1) Tuning-free techniques (Ku et al., 2024; Yang et al., 2025a) manipulate attention features to swap subjects, but struggle to restore fine details and largely ignore interactions with subjects; (2) Tuning-based methods inject the target subject either via text prompts like VideoPainter (Bian et al., 2025) or learned LoRA (Hu et al., 2022) from several subject concepts like VideoSwap (Gu et al., 2024); the former lacks detail fidelity, whereas the latter is computationally demanding and indirect in reference injection; (3) Recently, unified video customization frameworks (Jiang et al., 2025; Hu et al., 2025b) are emerging, aiming for versatility across multiple customization tasks, including video-driven subject swapping. Yet their all-in-one design sacrifices identity consistency and interaction realism in the subject swapping scenario.

To address these limitations, we present DreamSwapV, a mask-guided and subject-agnostic framework for video subject swapping. Given a source video, a user-specified mask localizing the source subject, and a reference image of the target subject, we can perform end-to-end swapping of any

---

*Corresponding Authors.
†Project Leader.

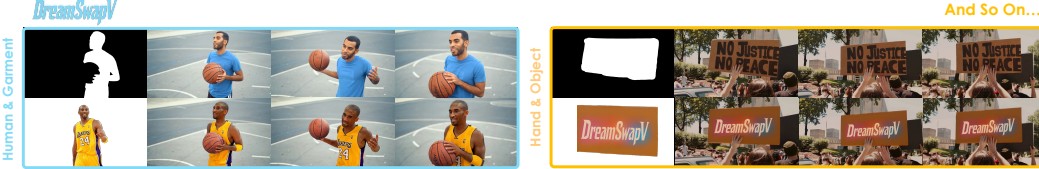

Figure 1: Our DreamSwapV swaps any subject in any video given a mask and reference image.

subject in any video. Unlike conventional editing paradigms that focus on *regenerating* or *editing* an external object into the scene, we recast swapping as an *inpainting* task: we encourage the model to *recover* the appearance of the target subject as if it inherently belonged within the missing masked region of the source video, yielding a direct and intuitive training and inference procedure.

Specifically, we first construct a task-oriented dataset based on HumanVID (Wang et al., 2024c) and introduce auxiliary conditions (mask, pose, 3D hand, etc.) for better video control. A dedicated condition fusion module then effectively integrates these conditions for the model's better understanding of both video content and task objectives. Furthermore, an adaptive mask strategy accommodates subjects of varying scales and attributes, mitigating shape leakage while enhancing subject-context interaction. Lastly, we establish DreamSwapV-Benchmark—the first benchmark tailored to video subject swapping—and evaluate our method along with 5 indicators inherited from VBench (Huang et al., 2024), confirming the robustness and superior performance of DreamSwapV.

Our contributions are summarized as follows:

- We propose DreamSwapV, the first end-to-end framework dedicated to the video subject swapping, specifically focusing on generic subject-context interaction.
- Our novel condition fusion module and adaptive mask strategy enable handling diverse subjects, enhancing contextual understanding, and improving video realism.
- Through a two-stage dataset construction and training scheme, we outperform existing methods on VBench indicators and first introduced DreamSwapV-Benchmark.

## 2 RELATED WORK

### 2.1 VIDEO GENERATION AND EDITING

Video generation is a pivotal component within the Artificial Intelligence Generated Content (AIGC) domain, realizing human creativity in the video medium. The field has undergone rapid evolution, from initial GAN-based methods (Vondrick et al., 2016; Tulyakov et al., 2018), through subsequent diffusion models (Yang et al., 2025c;b), to the current state-of-the-art Diffusion Transformers (DiT) (Peebles & Xie, 2023). At present, leading commercial models (Keling, 2025; Hailuo, 2025; Vidu, 2025) and open-source models (Yang et al., 2024; Kong et al., 2024) can leverage billion-scale architectures to generate high-fidelity videos at 720p-1080p resolutions.

Concurrently, editing techniques for real or generated videos are also advancing vigorously. Tune-A-Video (Wu et al., 2023) pioneers video editing with latent diffusion models via one-shot tuning, followed by (Qi et al., 2023; Cong et al., 2023; Yang et al., 2025d), enhancing capabilities in appearance detail, temporal consistency and motion control. With the advent of video foundation models, an increasing number of editing methods are based on video diffusion models (Gu et al., 2024; Yang et al., 2025a) or DiT (Bian et al., 2025; Tu et al., 2025), providing more precise and diverse video editing. Our DreamSwapV builds upon the cutting-edge Wan2.1-I2V-14B DiT model (Wan et al., 2025) and adapts it specifically to the video subject swapping task, with our condition fusion module injecting additional conditioning information and a carefully designed training scheme.

### 2.2 VIDEO CUSTOMIZATION

The freedom and creativity of personalized content creation facilitates the development of video customization, which can be broadly classified into two paradigms. **Subject concepts-driven methods** learn subject-specific adapters (Houlsby et al., 2019) or LoRA modules (Hu et al., 2022) from multiple reference images, embedding the subject concepts into the text space. DreamBooth (Ruiz et al., 2023) and Textual Inversion (Gal et al., 2022) pioneer this approach in the image domain and later extend to video customization by methods like (Chefer et al., 2024; Wu et al., 2025; Jiang et al.,

2024). While effective for identity preservation, the requirement of per-subject finetuning hampers real-time use, motivating the emergence of end-to-end methods.

**End-to-end customization** typically employs an additional conditioning network to inject user-specified information, similar to the idea of ControlNet (Zhang et al., 2023). Animation techniques (Hu et al., 2025a) design a ReferenceNet specifically for injecting detailed character appearance. Unified video customization frameworks (Jiang et al., 2025; Hu et al., 2025b) broaden the scope to encompass comprehensive customization capabilities, including text-to-video (T2V), reference-to-video (R2V), video-to-video (V2V) and masked-video-to-video (MV2V).

### 2.3 VIDEO SUBJECT SWAPPING AND INPAINTING

Video subject swapping is a derived sub-task of video customization, focusing on subject-driven editing and masked-video-to-video (MV2V), which has garnered significant attention due to rising user demand. Some general-purpose video editing methods (Ku et al., 2024; Yang et al., 2025a; Gu et al., 2024) inherently support video subject swapping, which operate in a *regenerating* or *editing* paradigm, rather than *video inpainting*.

Video inpainting (Xu et al., 2019; Zhou et al., 2023) itself is a long-established research field, originally concerned with restoring missing or damaged content in videos. Crucially, when the masked region represents an unknown external subject, video inpainting techniques can effectively *recover* this novel object, thereby achieving the functional goal of video subject swapping—as demonstrated by VideoPainter (Bian et al., 2025) and the MV2V capabilities of unified customization frameworks (Jiang et al., 2025; Hu et al., 2025b). Our DreamSwapV directly inherits and builds upon this core insight, by training a dedicated, end-to-end model specifically designed to achieve generic and high-quality video subject swapping in a video inpainting manner.

## 3 METHOD

### 3.1 TASK OVERVIEW

As fully discussed above, we cast video subject swapping as a special application of video inpainting: a user-specified mask $\mathbf{m}_0^s$ of the first frame delimits a region to be swapped, and a corresponding reference image $\mathbf{r}^s$ supplies the appearance of the new subject. Under this setting, the task can also transfer to (i) *standard video inpainting* when the reference is absent, or (ii) *video addition* when the masked region does not contain a pre-existing subject. In this work, we mainly focus on the canonical video subject swapping task—namely, the mask encloses a single subject and the reference depicts another subject of similar category.

**Data Filtering and Preprocessing.** To construct a dataset tailored for our task, we leverage HumanVID (Wang et al., 2024c) as the foundational resource. Beyond its human-centric focus, HumanVID offers extensive coverage of human-object interactions and diverse subject categories, aligning with our requirement for generic subject swapping.

Then we establish a rigorous data processing pipeline (video subject caption $\Rightarrow$ per-frame subject mask tracking $\Rightarrow$ quality filtering $\Rightarrow$ distribution standardization) to collect our final pre-training dataset. TikTok-VFM-7B is employed to caption prominent subjects in each video of HumanVID, followed by TrackingSAM (Cheng et al., 2023) performing per-frame tracking and segmentation of each subject. The obtained subject mask sequences then undergo strict quality filtering based on three criteria: *(i) the mask area ratio*, *(ii) the temporal coverage* (of total frames), and *(iii) the motion amplitude* (displacement across frames). To prevent our model from being overly biased towards any specific subject category, we further standardize the distribution of subjects in the dataset by filtering out excessively repetitive instances, ensuring a balanced diversity of subject categories.

Following these steps, we finally construct our pre-training dataset containing 8160 videos and 16219 subject instances (4701 *humans* : 1045 *garments* : 5477 *small objects* : 4996 *large objects* $\approx 1 : 0.2 : 1 : 1$, where *small objects* include highly human-related items like accessories and handheld objects, and *large objects* include other items like furniture, buildings, and background).

Pose and 3D hand sequences are detected together by DWPose (Yang et al., 2023) and Hamer (Pavlakos et al., 2024) per-frame and stored as auxiliary conditions required by Sec. 3.2. At last, each sample comprises (i) the raw video $\mathbf{V} = \{\mathbf{v}_t\}_{t=1}^T$, (ii) pose & 3D hand sequences $\mathbf{P} = \{\mathbf{p}_t\}_{t=1}^T$, and (iii) per-subject masks $\mathbf{M^s} = \{\mathbf{m}_t^s\}_{t=1}^T$, where $s$ means the specific subject.

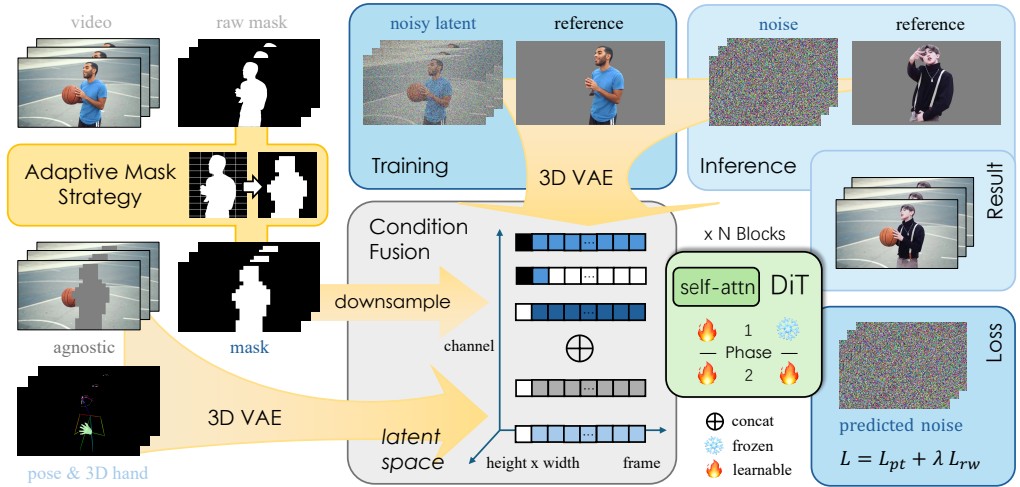

Figure 2: Overview of our framework. We first build a pre-training dataset containing video, raw mask, pose and 3D-hand sequences (Sec. 3.1). Our condition fusion module then integrates these signals with strict spatial–temporal alignment (Sec. 3.2). To improve subject-context interaction and enable general subject swapping, we introduce an adaptive mask strategy (Sec. 3.3). Finally, with a two-phase training scheme and additional technical implementations (Sec. 3.4), we deliver state-of-the-art subject-swapping results at inference time.

**Training and Inference.** During training, the reference image is collected by $\mathbf{r}' = \mathbf{v}_i \odot \mathbf{m}_i$, where $i$ is a randomly selected frame and $\odot$ denotes element-wise product, i.e., extracting the subject marked by the mask from the video frame as the training reference image. The model is thus trained to *recover* the original video by inpainting the masked region using this extracted reference image, encouraging faithful appearance recovery and seamless interaction with the scene, which can be expressed as follows:

$$\text{Loss} = L_{pt}(\mathbf{V}, \mathbf{V}'), \mathbf{V}' = f_\theta(\mathbf{M^s}, \mathbf{P}, \mathbf{r}') \tag{1}$$

where $L_{pt}$ represents the pre-trained loss, whose final calculation will be supplemented in Sec. 3.4, and $f_\theta$ depicts the *recovering* procedure of our model with weight $\theta$.

At inference time, the model treats an external reference image $\mathbf{r^s}$ as if it were the subject originally extracted from the mask region, leveraging *recovering* capability learned during training. Thus we can obtain the customized video $\mathbf{V}'$ after our model's subject swapping: $\mathbf{V}' = f_\theta(\mathbf{M^s}, \mathbf{P}, \mathbf{r^s})$. The whole process is overviewed in Figure 2.

## 3.2 CONDITION FUSION MODULE

The subject swapping task inherently requires the model to distinguish between regions requiring to be swapped and those to be preserved, which is relatively straightforward to learn with the binary mask serves as a key condition. Building on this, the more challenging issues are (i) accurately recovering the motion trajectory of the swapped subject, and (ii) handling the mask boundary interactions between the swapped subject and its surrounding context.

For dynamic subjects (e.g., humans and animals), self-occlusion frequently occurs during movement, necessitating the pose estimation as an additional condition to provide the subject's motion information. For static subjects (e.g., objects), the motion pattern from camera movement or external forces is simpler, and usually can be inferred from mask shape changes. However, the realism of interactions between these objects and external forces (e.g. hand manipulations) also critically impacts visual quality. To better handle these interactions, we introduce 3D hand estimation as a complementary signal, providing hand-object interaction information. The pose and 3D hand estimation are combined as a unified temporal sequence $\mathbf{P}$ to offer their auxiliary control.

Along with the necessary mask sequence $\mathbf{M^s}$, the masked video sequence $\mathbf{A^s} = \mathbf{V} \odot (\mathbf{1} - \mathbf{M^s})$ (often referred to as agnostic) and the reference image $\mathbf{r}^s$, we incorporate four additional conditions along with the noisy video input. Efficiently coupling this multi-modal information is essential to overall performance, where we address this challenge through a dedicated condition fusion module.

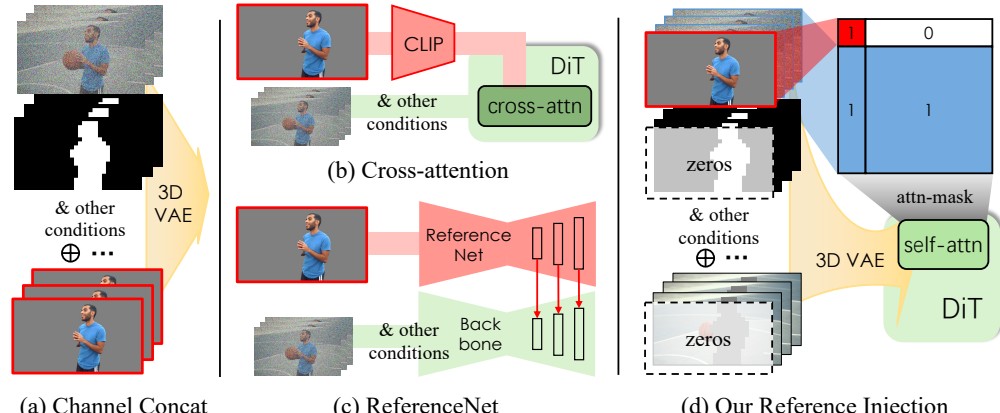

Figure 3: Schematic comparison between reference injection alternatives (a)-(c) and ours (d).

**Latent Space Projection.** All input sequences except the binary masks are encoded into a shared latent space using a pre-trained 3D VAE. The shape of these latents becomes [batch, frame, channel, height, width] = [b, f, c, h, w], with their temporal dimension [f] compressed by 4 and spatial dimensions [h, w] compressed by 8. The reference latent remains f = 1, and the remaining sequences are scaled to f = (frames - 1) // 4 + 1. Next, the first frame of the noisy video latent is extracted separately to form a dummy reference latent, with its subsequent frames zero-padded to match the temporal length of the noisy video latent. This latent explicitly represents the first frame of the whole video sequence, enabling the model to focus on the extension of the first frame, which provides an interface for extrapolating the video length and first frame reference as detailed in Sec. 3.4.

**Reference Information Injection.** We concatenate the reference latent along the temporal dimension [f] with both the noisy and dummy reference latents, and compare with alternatives below:

(i) *Channel concatenation:* It is a straightforward idea to concatenate the zero-padded or all-copied reference latent along the channel dimension, just as we do with the dummy reference latent and other latents. However, this may disrupt the spatial-temporal feature alignment across frames since the reference latent represents a global signal applicable to all frames, not just frame 0, leading to confused learning and impaired detail injection.

(ii) *Cross-attention:* Extracting features such as CLIP embeddings instead of 3D VAE latents for cross-attention leads to limited fine-detail capture due to encoder bottlenecks. Also, cross-attention is better suited for semantic guidance (e.g., text embeddings), not high-fidelity appearance injection.

(iii) *ReferenceNet:* In animation methods, an additional ReferenceNet is typically used to implicitly inject reference features. However, as demonstrated by UniAnimate (Wang et al., 2024a), such operation introduces parameter redundancy and feature space misalignment between the reference image and noisy video latent, which can be replaced.

Therefore, we decide to use frame-level concatenation, extending the token length and enabling seamless reference injection. Since the reference is only used for feature extraction and not for prediction, in the self-attention mechanism, the video attends to the reference while the reference only attends to itself (using KV cache), as shown in the attention mask of Figure 3. The reference latent is also excluded from the loss computation, thereby serving as an equivalent implementation to ReferenceNet in a more concise temporal concatenation manner.

**Mask Conditioning and Final Fusion.** The binary mask sequence does not require 3D VAE encoding. To match the spatial-temporal dimensions of the other condition latents, the masks are processed by concatenating every four frames along the channel dimension, followed by an 8x downsampling in spatial dimensions, resulting in a [b, (f - 1) // 4 + 1, 4, h // 8, w // 8] shape. Finally, the mask, agnostic latent, and pose latent are zero-padded on the reference frame, and all the latents are concatenated along the channel dimension to fuse as the final **model_input**.

In summary, our condition fusion module ensures strict spatial-temporal alignment across all conditioning signals. For each frame in the temporal dimension, corresponding conditions of the frame are concatenated along the channel, with additional reference information injected globally via token extension, simplifying the model's learning.

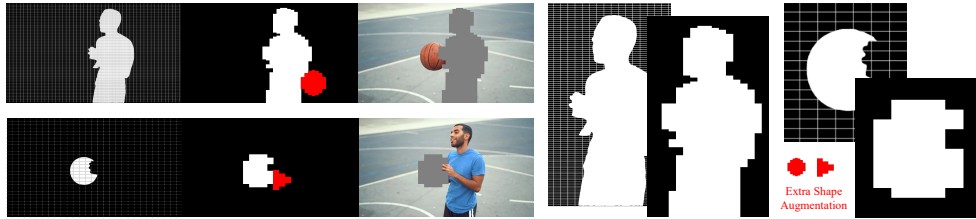

Figure 4: Illustration of adaptive grid sizing across subject scales with extra-shape augmentation.

## 3.3 ADAPTIVE MASK STRATEGY

As a mask-guided task, the handling of the mask is crucial for video subject swapping. *Overly precise* masks can cause the model to overfit to the mask shape during training, leading to poor generalization (commonly known as *shape leakage*) when swapping subjects with large domain gaps (e.g. square box ⇒ spherical ball). Conversely, *overly coarse* masks may confuse the model's learning, producing artifacts and less refined details. Proper treatment of the mask boundary is therefore essential, governing the subject-context interaction and visual quality of the final result.

Animate Anyone 2 (Hu et al., 2025a) addressed such dilemma using a grid-based mask augmentation strategy: the bounding box of the mask is divided into $k_h \times k_w$ non-overlapping blocks; any block containing mask pixels is dilated, effectively blurring the precise mask boundary to reduce *shape leakage*. While Animate Anyone 2 is designed specifically for human swapping (animation), our goal is to perform generic subject swapping across diverse scales and attributes. Directly applying their augmentation strategy would result in both small items (e.g. accessories, handheld objects) and relatively large objects (e.g. humans, vehicle, background) sharing the same augmentation, which is suboptimal for both. Thus, we need to explore a more generalizable mask augmentation strategy.

**Adaptive Grid Sizing.** To begin with, we randomly use a bounding box augmentation with 30% probability, representing the largest shape augmentation (coarsest mask). In the remaining 70% of cases, we inherit the grid-augmentation strategy of Animate Anyone 2, dividing the whole frame rather than the bounding box and making the grid size inversely proportional to the subject scale:

$$\underbrace{\begin{cases} k_h^{train} = \mathrm{rand}(1, h) \\ k_h^{inf} = h//10 \end{cases}}_{\text{Animate Anyone 2}} \Rightarrow \underbrace{\begin{cases} K_h^{train} = \mathrm{bbox\_h}//\mathrm{rand}(h_1, h_2) \\ K_h^{inf} = \mathrm{bbox\_h}//h_3 \end{cases}}_{\text{Our Adaptive Grid Sizing}} \quad (2)$$

where $k_h^{train}, k_h^{inf}$ denote the number of vertical blocks into which Animate Anyone 2 divides the mask's bounding box during training and inference; $K_h^{train}, K_h^{inf}$ calculate our adaptive vertical block number, based on the bounding box height $\mathrm{bbox\_h}$ and hyper-parameters $h_1, h_2, h_3$.

The horizontal counts $k_w, K_w$ are computed analogously. Intuitively, the larger the subject, the larger $\mathrm{bbox\_h} \times \mathrm{bbox\_w}$, and hence the larger $K_h \times K_w$, which yields a finer grid, better controlling their movement (and the larger subjects' swapping rarely involves *shape leakage*). Conversely, small subjects receive coarser masks which accommodate more diverse swapping (e.g., handheld objects to other categories). We empirically find that this adaptive strategy improves the generalization for small subjects and enhances the edge and motion control over larger subjects in ablation studies.

**Extra Shape Augmentation.** After mask augmentation, it can be observed that the augmented mask is consistently larger than the target subject. Animate Anyone 2 considers this problematic, arguing that the model may develop a bias in which the generated subject is always smaller than the provided mask; thus, they introduce a random scale augmentation. However, in our view, as long as training and inference follow the same mode—namely, the mask is always slightly larger than the subject—this does not introduce a harmful bias. Instead, it simplifies the objective: the model only needs to generate the subject within a mildly expanded region rather than infer a precise mask–subject correspondence. Therefore, we retain this as the default setting.

This design, however, means that the augmented mask inevitably covers some background, requiring the model to perform non-subject background completion. Without additional constraints, the model tends to fill the entire mask with reference-derived content, leading to implausible hallucinations, such as extending extra hair into the masked region rather than filling reasonable background. To

mitigate this, we introduce extra-shape augmentation during training. At random intervals, we add simple geometric shapes (circle, triangle, rectangle) to the edges of the grid-augmented mask. This design further decouples the subject from the mask's exact shape, teaching the model that not all masked pixels belong to the subject. Consequently, the model learns to fill background content more appropriately along the mask boundary, improving subject-context interaction.

Overall, our adaptive mask strategy forms a generic and scalable augmentation scheme that adapts to subjects of varying sizes and attributes. Together with the pose and 3D-hand conditions, the grid-based and extra shape augmentations provide richer guidance, producing smoother, more natural boundaries between the swapped subject and its surroundings.

### 3.4 Implementation Details

We list several technical details that, while not totally novel, proved essential to final performance.

**Mask Constraint.** Since our model is strictly trained in a mask-guided manner, the accuracy of the mask is crucial for final performance. Any regions not covered by the mask will be strictly preserved and serve as context. If the mask contains certain inaccuracies that leak parts of the source subject, the model may draw cues from these regions during swapping, potentially reconstructing the source subject rather than injecting the external reference.

To mitigate this, besides providing a precise mask sequence rather than relying on TrackingSAM, we recommend a bounding-box inference mode, which expands the mask to bounding-box to fully cover the source subject. Moreover, fast-moving or heavily occluded objects may cause TrackingSAM to produce inconsistent masks. At inference, we assume consecutive masks share some spatial overlap; if two adjacent masks overlap $\leq 5\%$, we use their minimal enclosing region to ensure coverage.

**Reference Augmentation.** During pre-training, the reference image is always extracted from the same video, maintaining its perfect scale and luminance. This risks the model learning a trivial *copy-paste* behavior (also noted in HunyuanCustom), impairing its ability to handle real-world references with scale, brightness, or angle mismatches. To mitigate this, we apply random scaling, rotation, flipping, and brightness adjustment to the training reference. For severe domain gaps (e.g., inserting a half-body human into a full-body mask), we experiment with random cropping, which reduces the gap but introduces new hallucinations and detail loss. As a trade-off, we disable such cropping.

**Two-phase Training Scheme.** As mentioned above, the pre-training dataset only uses references extracted from their source videos, limiting generalization to external references with deformations, brightness discrepancies or domain gaps. We address this with a two-phase training:

(i) *Pre-training (self-attention only):* We first train the base model on our HumanVID-derived dataset (Sec. 3.2) with only the self-attention layers trainable, preserving the base model's foundational generative capabilities. Each reference-video pair shares the same domain in this phase.

(ii) *Quality tuning (full fine-tuning):* We assemble a smaller, high-quality dataset of diverse reference-video pairs by (a) extracting paired images from AnyInsertion (Song et al., 2025) and Subject200K (Tan et al., 2024), and converting one image of each pair to video via Wan2.1-I2V (Wan et al., 2025), and (b) adding ∼400 handheld reference-video pairs from AnchorCrafter-400 (Xu et al., 2024c). All the model layers are unfrozen for full fine-tuning, enabling better convergence and adaptation to cross-domain reference-video pairs.

**Length Extension and First Frame Reference.** Our design (Sec. 3.2, dummy reference latent) provides an interface for handling long videos and leveraging first-frame priors.

(i) *Long video processing:* For videos longer than our training length, we first split the video sequence into overlapping segments: the last frame of the previous segment becomes the first frame and the dummy reference for the next. In subject swapping, the majority of scene content remains constant between segments. This inherent context enables stable temporal extrapolation without the disruptive scene jumps common in general image-to-video tasks.

(ii) *First frame reference:* Since the dummy reference concentrates the model's attention on the first frame, any edit applied there propagates coherently. At inference, users may optionally supply a first-frame swap produced by an image subject swapping model like (Chen et al., 2024), enhancing visual fidelity. Notably, DreamSwapV is inherently compatible with single frame swapping, so leveraging specialized image-based models is an alternative option for extra stability.

**Tunnel Video Inpainting and Reweighting Loss.** When the mask is extremely small (e.g., accessory), the model may overlook fine details. Inspired by Tunnel Try-on (Xu et al., 2024a) and AnchorCrafter (Xu et al., 2024c), we tackle this issue in parallel during both inference and training:

(i) *Inference—tunnel inpainting:* At inference time, when the mask area ratio falls below the predefined threshold (0.05), we extract a tight crop (*tunnel*) around the mask, perform subject swapping in this sub-region, and blend this tunnel region back into the full frame, which can focus on pixels where it matters, improving detail fidelity on tiny objects.

(ii) *Training—reweighting loss:* Following AnchorCrafter(Xu et al., 2024c), we enhance the model's attention to small subject regions by introducing a subject-region reweighting loss:

$$L_{\text{rw}} = \frac{\mathbf{E}}{\mathbf{E}^s} \mathbf{M}^s \odot L_{\text{pt}}, \ L_{\text{final}} = \left(1 - \mathbf{M}^s\right) \odot L_{\text{pt}} \ + \ \lambda L_{\text{rw}} \tag{3}$$

where $L_{\text{pt}}, L_{\text{final}}, L_{\text{rw}}$ are pre-trained loss, overall training objective, and re-weighting loss that amplifies the learning signal inside the subject mask region, respectively; $\mathbf{E}$ and $\mathbf{E}^s$ are the areas of the whole frame and the subject, and $\mathbf{M}^s \in \{0, 1\}^{h \times w}$ indicates the subject binary mask.

## 4 EXPERIMENTS

### 4.1 BASELINES AND BENCHMARK

**Basic Information.** Our DreamSwapV is trained on the Wan2.1-I2V-14B foundation model, supporting up to 720p resolution. Our training follows a two-phase scheme: pre-training for 15000 iterations and quality-tuning for 10000 iterations, both on 32 NVIDIA H100 80GB GPUs. Each 100 iterations cost approximately 75 minutes, yielding a total training duration of ~13 days. Notably, since we do not modify or depend on the model architecture, our entire framework can be readily transferred to other base models, such as subsequent versions of the WanX series(Wan et al., 2025) or alternatives like CogVideoX(Yang et al., 2024) and HunyuanVideo(Kong et al., 2024).

**Benchmark.** While VBench series (Huang et al., 2024) have provided valuable benchmarks for text-to-video and image-to-video tasks, similar evaluations for video subject swapping remain absent. To address this gap, we introduce DreamSwapV-Benchmark, the first benchmark for video subject swapping. Following the scale of benchmarks used in VACE and HunyuanCustom, we collect 100 videos from the online website (Pexels, 2025), and annotate 167 distinct subject instances from them. Original videos cover four aspect ratios to test cross-resolution generalization. For each subject, we segment and track its precise mask sequence in the source video, and match it with an appropriate reference image for swapping. The subjects and their reference images for swapping are carefully selected to ensure a wide and balanced distribution across categories and complexities.

We adopt 5 indicators inherited from VBench (see the top 5 metrics in Table 1) and design 3 other automatic metrics specifically for the subject swapping task: reference appearance, background preservation and semantic consistency. We also introduce an user study to evaluate human preferences for three aspect: reference detail, subject interaction and visual fidelity. The full benchmark distribution, metric calculation rules and user study details are provided in Appendix C.2.

**Baselines.** We compare our DreamSwapV against 3 open-source methods (AnyV2V, VACE, HunyuanCustom) and 1 commercial model (Kling 1.6 Multimodal). We select them for their robust stability, highly accessiblity, and close relevance to our subject swapping objective. We exclude text-instruction based methods (VideoGrain, VideoSwap, and VideoPainter) to avoid unfair comparisons under distinct task settings. See Appendix C.3 for detailed baseline implementations.

### 4.2 EVALUATION

**Quantitative Comparison.** Table 1 reports the quantitative results on DreamSwapV-Benchmark. Across the five VBench indicators, our DreamSwapV achieves the best average score, marginally surpassing Kling 1.6. The gap widens when Kling 1.6 suffers from its *regeneration-like* framework, altering large portions of the background—or even redrawing entire frames—compromising its practical usability. VACE and HunyuanCustom underperform mainly in reference appearance and detail consistency, sometimes failing to insert the correct target subject due to their unified task focus, where subject swapping is only one objective. AnyV2V's unstable intermediate feature manipulations often lead to complete video collapse, severely impairing visual fidelity. Overall, DreamSwapV delivers the most stable and high-quality results, corroborated by the user study.

Table 1: Comparison of video subject swapping methods on DreamSwapV-Benchmark. The **bold** and underline are the best and second-best results, respectively, and the gray represents our results.

| Method / Metrics | Video Quality & Video Consistency | | | | | | | | | | User Study | | |
| --- | --- | --- | --- | --- | --- | --- | --- | --- | --- | --- | --- | --- | --- |
| | Subject Consistency | Background Consistency | Motion Smoothness | Dynamic Degree | Aesthetic Quality | VBench Average | Reference Appearance | Background Preservation | Semantic Consistency | Total Average | Reference Detail | Subject Interaction | Visual Fidelity |
| AnyV2V | 90.03% | 91.35% | 98.60% | 47.90% | 51.79% | 75.93% | 34.70% | 42.71% | 51.00% | 63.51% | 0.87 | 0.65 | 0.42 |
| VACE | 96.15% | 95.03% | 99.29% | 27.54% | 56.95% | 74.99% | 39.66% | 47.46% | 66.93% | 66.16% | 2.09 | 2.31 | 2.46 |
| HunyuanCustom | 95.83% | 94.96% | 99.17% | 43.11% | **57.78%** | 78.17% | 41.33% | 48.14% | 63.65% | 68.00% | 2.17 | 2.22 | 2.13 |
| Kling 1.6 | 95.36% | **96.57%** | **99.45%** | 50.33% | 57.26% | 79.79% | 42.27% | 39.17% | 69.95% | 68.80% | 3.04 | 2.89 | 3.14 |
| **DreamSwapV** | **96.41%** | 94.26% | 99.31% | 55.69% | 56.52% | **80.44%** | **45.22%** | 52.49% | **72.01%** | **71.49%** | **3.35** | **3.39** | **3.32** |

Figure 5: Qualitative comparisons between our DreamSwapV and other baselines on various subjects and video aspect ratios. Please zoom in for details, and refer to supplementary dynamic videos.

**Qualitative Results.** Figure 5 provides a visual comparison highlighting DreamSwapV's advantages across 6 different kinds of subjects (see Appendix D for more results). In the *Human⇒Luffy* case, DreamSwapV successfully swaps the person with Luffy, preserving his original pose and reference details, while Kling struggles with reference consistency, HunyuanCustom and VACE inject an

Table 2: Quantitative comparison of different ablation settings (w/o reference injection, w/o adaptive grid sizing, w/o extra shape augmentation or w/o two-phase training) with our full version model. The **bold** are the best results under certain metrics, and gray marks the full version model.

| Ablation / Metrics | Video Quality & Video Consistency | | | | | | | | | |
|---|---|---|---|---|---|---|---|---|---|---|
| | Subject Consistency | Background Consistency | Motion Smoothness | Dynamic Degree | Aesthetic Quality | VBench Average | Reference Appearance | Background Preservation | Semantic Consistency | Total Average |
| w/o our reference injection | | | | | | | | | | |
| *- channel concat* | 93.87% | 93.24% | 99.11% | 48.60% | 54.17% | 77.80% | 37.34% | 51.57% | 58.22% | 67.15% |
| *- cross-attention* | 95.49% | 94.21% | 99.29% | 52.67% | 56.41% | 79.61% | 39.23% | 51.84% | 62.93% | 69.11% |
| w/o our adaptive grid sizing | 96.07% | 94.24% | 99.31% | 54.37% | 56.39% | 80.08% | 39.29% | **52.70%** | 65.55% | 69.81% |
| **w/o our extra mask aug.** | 96.12% | 94.11% | 99.30% | 55.66% | 55.69% | 80.19% | 43.61% | 52.12% | 66.03% | 70.34% |
| **w/o our two-phase training** | 96.38% | 94.04% | 99.31% | 55.67% | 55.37% | 80.15% | 43.52% | 52.41% | 65.14% | 70.23% |
| **Full (ours)** | **96.41%** | **94.26%** | 99.31% | **55.69%** | **56.52%** | **80.44%** | **45.22%** | 52.49% | **72.01%** | **71.49%** |

Figure 6: Ablation examples on our reference injection and adaptive grid sizing, zoom in for details.

almost static subject, and AnyV2V collapses completely. In the *Coat⇒Coat* and *Pen⇒Pocky* case, all baselines fail to maintain the reference appearance, whereas DreamSwapV retains fine details and realistic motion, yielding the highest visual quality.

**More applications.** It is worth noting that DreamSwapV is not limited to video subject swapping, but can extend to several related applications like *image swapping, video inpainting, video addition, video try-on* and so on. We further discuss these extensions in Appendix E.

**Limitations and failure modes.** As discussed in Section 3.4, certain cross-domain swapping modes (e.g., inserting a half-body human into a full-body mask) pose challenges for our model. For cases such as animal → human or object → character, the poses of animals or objects differ significantly from humans, further increasing the difficulty for our pose-conditioned approach. We leave the task of cross-domain swapping to future mask-free work. For more limitations and our future plan, please refer to Appendix E.4.

### 4.3 ABLATION STUDY

We ablate on our reference injection of condition fusion module and adaptive grid sizing with visual examples in Figure 6. Our reference injection achieves finest details over alternatives like *channel concatenation* and *cross-attention*. Our adaptive grid sizing can better handle subjects with different scales and improve visual fidelity. We also conduct quantitative ablation study on reference injection, adaptive grid sizing, extra shape augmentation and two-phase training in Table 2.

## 5 CONCLUSION

In this paper, we present DreamSwapV, a mask-guided and subject-agnostic framework for end-to-end, generic subject swapping in video customization. We introduce a dedicated condition fusion module and an adaptive mask strategy to integrate multiple conditioning signals efficiently, enabling fine-grained control and more natural subject-context interactions. Our elaborate two-phase dataset construction and training scheme further enhance DreamSwapV's capabilities, enabling it to outperform all existing methods on VBench indicators and our first introduced DreamSwapV-Benchmark.

## ACKNOWLEDGEMENTS

This work is supported by the NSFC fund (62576190), in part by the Shenzhen Science and Technology Project under Grant (KJZD20240903103210014, JCYJ20220818101001004)

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

## A  OVERVIEW

This technical appendix includes additional details and information about our methods and experiments which cannot be fully covered in the main paper limited by space. We first strongly recommend readers to watch the video demo (Appendix B) provided along with this appendix in the supplementary material, where we dynamically showcase our entire pipeline and present the video comparison results. Then we complement our full implementation details for reproducibility, including hyper-parameters, benchmark contents, and baseline adaptations (Appendix C). We further provide additional experimental results which cannot be fully displayed in the main paper(Appendix D). At last, we provide a comprehensive discussion, including the more applications, social impact, large language model (LLM) usage, limitations and future work (Appendix E).

## B  VIDEO DEMO

Our video demo is provided in the supplementary material and consists of:

- The brief presentation summarizing the main paper.
- Dynamic pipeline and data flow, which brings a clearer and more complete understanding of our proposed method.
- Video comparisons of our method and four baselines across diverse subject categories and aspect ratios.
- More application examples demonstrating the strong extensibility of our method.

It is worth noting that, due to the **100 MB file size limit**, the video demo has been compressed and may appear slightly degraded in quality. The original results are clearer and smoother; however, since all videos are compressed to the same degree, the relative performance among methods remains unaffected.

## C  IMPLEMENTATION DETAILS FOR REPRODUCIBILITY

### C.1  SPECIFIC HYPER-PARAMETERS

The specific hyper-parameters of DreamSwapV in our two-phase training scheme are listed in Table 3 for better reproducibility.

### C.2  DREAMSWAPV-BENCHMARK CONTENTS

**Benchmark Distribution.**    The subjects and their reference images for swapping from DreamSwapV-Benchmark are carefully selected to ensure a wide and balanced distribution across categories and complexities.

Figure 7a is the 2-D histogram density of all reference images after t-SNE (Maaten & Hinton, 2008) projection: each colour block represents the number of images falling into a $8 \times 8$ grid map on the embedding plane. The cell-count coefficient of variation is 0.736 and the $\chi^2$ goodness-of-fit test gives p = 0.952 (> 0.05), showing that only a very small fraction of grid cells are empty or over-populated—i.e., the images cover the feature space almost uniformly.

Figure 7b is the t-SNE scatter plot coloured by K-Means clusters (k = 12) produced in the original 2048-D feature space. The clusters have a size CV of 0.495 and a normalised Shannon entropy of 0.947 ($\sim 1$), meaning the images are evenly split across latent semantic groups with no dominant or under-represented cluster.

Together, the two figures verify that our benchmark set is both spatially well-covered and semantically balanced, preventing bias in subsequent experiments.

**Metric Calculation.** We adopt five indicators inherited from VBench (Huang et al., 2024)—subject consistency, background consistency, dynamic degree, motion smoothness, and aesthetic quality—and design two other automatic metrics specifically for the subject swapping task: reference

Table 3: Hyper-parameters in our two-phase training.

| Hyper-parameters | Pre-training |
|---|---|
| Iterations | 15000 |
| Learning Rate | 2e-5 |
| LR Scheduler | `constant_with_warmup` |
| Batch size | 1 |
| Optimizer | AdamW |
| Resolution | 1280×720 |
| Frame Length | 65 |
| CFG Drop Ratio | 0.15 |
| Image Mix Ratio | 0.3 |
| Flow Shift | 5.0 |
| $h_1, h_2, h_3$ | 1, 30, 20 |
| | **Quality Tuning** |
| Iterations | 10000 |
| Learning Rate | 1e-5 |
| LR Scheduler | `constant_with_warmup` |
| Batch size | 1 |
| Optimizer | AdamW |
| Resolution | 1280×720, 720×1280, 512×512, 1024×1024 |
| Frame Length | 69 |
| CFG Drop Ratio | 0.15 |
| Image Mix Ratio | 0.1 |
| Flow Shift | 5.0 |
| $h_1, h_2, h_3$ | 1, 30, 20 |

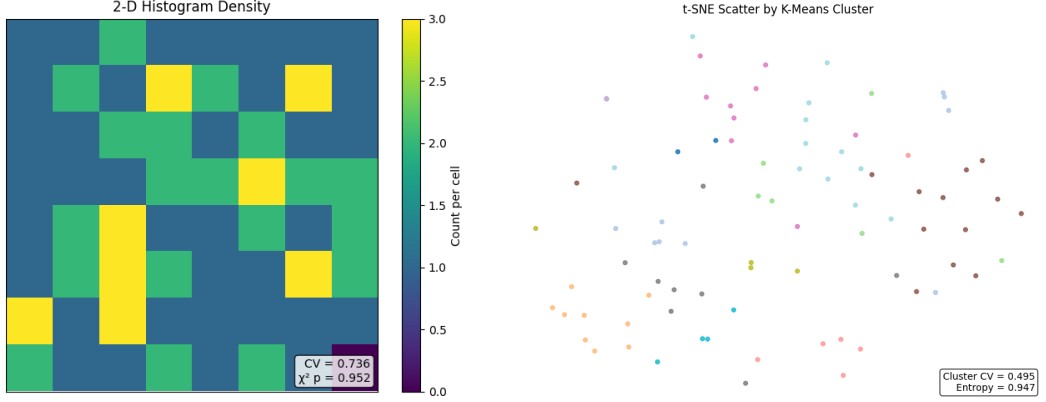

(a) 2-D histogram density of references (t-SNE plane).     (b) t-SNE scatter coloured by K-Means (k = 12).

Figure 7: DreamSwapV-Benchmark distribution analysis. (a) shows spatial coverage and (b) shows semantic balance across clusters.

appearance and background preservation. The specific meanings and calculations of each metric are listed below:

*(i) Subject consistency:* The indicator measures temporal subject-identity consistency—i.e., how stable the visual appearance of the main subject remains from frame to frame within a single video. Let $d_t \in \mathbb{R}^D$ denote the $\ell_2$-normalised DINO (Caron et al., 2021) feature of frame $t$ in a $T$-frame video. Temporal subject identity stability is quantified by

$$S_{\text{subject}} = \frac{1}{T-1} \sum_{t=2}^{T} \frac{\langle d_1, d_t \rangle + \langle d_{t-1}, d_t \rangle}{2},$$

where $\langle \cdot, \cdot \rangle$ is the cosine similarity. The score is averaged over all test videos of a model; larger values indicate greater subject consistency.

*(ii) Background consistency:* This indicator measures the temporal consistency of the *background identity* throughout a video. For each of the $T$ frames, a CLIP image encoder (Radford et al., 2021) yields a unit-normalised feature vector $c_t$. Temporal background consistency is then

$$S_{\text{background}} = \frac{1}{T-1} \sum_{t=2}^{T} \frac{\langle c_1, c_t \rangle + \langle c_{t-1}, c_t \rangle}{2},$$

where $\langle \cdot, \cdot \rangle$ denotes cosine similarity. The model-level score is obtained by averaging $S_{\text{background}}$ across all evaluation videos; larger values indicate a more consistent background.

*(iii) Dynamic degree:* This dimension gauges the tendency of a model to produce *non-static* videos. For each generated clip $v$ with $T$ frames, RAFT (Teed & Deng, 2020) is employed to estimate dense optical flow between consecutive frames, yielding magnitudes $\{m_{v,t}\}_{t=1}^{T-1}$. Let $\lambda_v = \text{mean}(\text{top-5}\%\{ m_{v,t} \})$ denote the average of the largest $5\%$ flow magnitudes, which is sensitive to small-object motion. A clip is classified as *dynamic* if $\lambda_v > \tau$, where $\tau$ is a fixed empirical threshold. The model's dynamic degree is the proportion of dynamic clips:

$$S_{\text{dynamic}} = \frac{1}{N} \sum_{v=1}^{N} \mathbf{1}[\lambda_v > \tau],$$

with $N$ the number of evaluation videos and $\mathbf{1}[\cdot]$ the indicator function. Higher values imply a stronger propensity to generate motion.

*(iv) Motion smoothness:* Smooth, physically plausible motion is assessed via a frame-interpolation prior. Given a $(2n+1)$-frame clip $\{f_t\}_{t=0}^{2n}$, every second frame is removed to form the low–frame-rate sequence $\{f_{2k}\}_{k=0}^{n}$; a pretrained interpolator then predicts the missing frames $\{\hat{f}_{2k+1}\}_{k=0}^{n-1}$. Temporal smoothness is quantified by the complement of the normalised mean absolute error (MAE) between the predicted and original odd frames:

$$S_{\text{motion}} = 1 - \frac{1}{n} \sum_{k=0}^{n-1} \frac{\|f_{2k+1} - \hat{f}_{2k+1}\|_1}{C},$$

where $C$ is the normalisation constant (as in Eq. 4) that bounds $S_{\text{motion}} \in [0,1]$. Averaging $S_{\text{motion}}$ across all test videos yields the model-level score; larger values indicate smoother motion.

*(v) Aesthetic quality:* Photographic composition, colour harmony, and general artistic appeal are assessed with the LAION image-aesthetics predictor (LAION-AI, 2022), which returns a raw per-frame score $a_t \in [0, 10]$ for each of the $T$ frames. Scores are linearly mapped to $[0, 1]$ and averaged:

$$S_{\text{aesthetic}} = \frac{1}{T} \sum_{t=1}^{T} \frac{a_t}{10}.$$

Averaging $S_{\text{aesthetic}}$ across all clips gives the model-level score; larger values indicate higher aesthetic quality.

*(vi) Reference appearance:* While the VBench *subject consistency* metric verifies that the swapped subject remains visually stable across video frames, it does not focus how faithfully that subject

mirrors the *external reference image* provided by the user. We therefore introduce the reference appearance metric to measure appearance fidelity. The same DINO backbone used in *subject consistency* is employed, but the calculation is changed: each video frame is compared directly to the reference image rather than to other frames.

Let $d_t$ be the $\ell_2$-normalised DINO feature of frame $t$ in a $T$-frame clip and $d_r$ the feature of the reference image. The metric is the mean cosine similarity

$$S_{\text{reference}} = \frac{1}{T} \sum_{t=1}^{T} \langle d_r, d_t \rangle,$$

which lies in $[0, 1]$. Higher values indicate that the swapped subject in the video retains the color, texture, and overall appearance of the reference image more closely. By complementing subject-consistency, this metric allows a comprehensive assessment of both temporal stability and appearance fidelity in video subject-swapping tasks.

*(vii) Background preservation:* The VBench *background-consistency* metric measures intra-video background identity but cannot reveal how much the scene diverges from the *original* footage after video subject swapping. To capture that difference, we supply with the background preservation metric directly comparing each swapped frame to its source counterpart outside the tracked object mask.

For every time step $t \in \{1, \ldots, T\}$, let $U_t$ and $V_t$ be the original and swapped frames, and $\mathbf{M}_t^s$ denote the background pixels (the complement of the user-provided mask). The peak signal-to-noise ratio on the background region is $P_t = \text{PSNR}(U_t \odot \mathbf{M}_t^s, V_t \odot \mathbf{M}_t^s)$ (measured in dB). A monotone rescaling maps PSNR to $(0, 1)$ and is averaged across the video frames:

$$S_{\text{preservation}} = \frac{1}{T} \sum_{t=1}^{T} \frac{P_t/50}{P_t/50 + 0.6}.$$

Higher values indicate that the background is better preserved after swapping the subject, whereas lower scores signal unintended scene alterations.

*(viii) Semantic consistency:* Subject swapping may change the high-level semantics of the target subject. To assess whether the generated video preserves the semantic meaning specified by the reference, we compute a caption-based semantic similarity using BLIP-2(Li et al., 2023), a recent vision-language model capable of producing robust semantic embeddings.

For each time step $t \in \{1, \ldots, T\}$, BLIP-2 generates a caption embedding $c_t$ from the masked subject region in frame $t$, and $c_r$ denotes the caption embedding extracted from the reference image. All embeddings are $\ell_2$-normalized. The semantic consistency score is defined as the mean cosine similarity:

$$S_{\text{semantic}} = \frac{1}{T} \sum_{t=1}^{T} \frac{1}{1 + \exp(-k\, s_t)},$$

where $k = 50$ is a fixed parameter that maps the small raw similarities into a stable $(0, 1)$ range while preserving their ordering.

This metric reflects whether the video maintains the category, coarse shape, and other high-level semantics of the reference subject, thereby complementing appearance- and background-based evaluations.

**User Study Details.** To further examine human preferences of our method and four baselines, we conduct a user study from three aspects:

- *Reference detail* instructs annotators to judge how faithfully the fine-grained details of the reference image are retained after subject swapping, paralleling the automatic metric *reference appearance*.
- *Subject interaction* focuses on the interaction between the inserted subject and its surroundings, assessing both the visual plausibility of that interaction and whether it remains strictly confined to the user-specified region.

Figure 8: The streamlit annotation interface of user study.

- *Visual fidelity* gauges the overall perceptual quality of the video—how realistic it appears and whether any synthetic or *AI-generated* artifacts are discernible.

15 annotators are carefully selected: five specialists in video generation or editing, five practitioners or students from other areas of artificial intelligence (AI), and five individuals with no AI background. The streamlit (Khorasani et al., 2022) annotation interface (Figure 8) presents each rater with five videos at a time. For each metric, the annotators are required to produce a strict ranking of the five videos from first to fifth, with no ties permitted.

The rankings are then converted into numerical scores using a weighted Borda-count (Emerson, 2013) scheme: 1st = 5 points, 2nd = 3 points, 3rd = 2 points, 4th = 1 point, and 5th = 0 points. Scores are aggregated per metric and per method, and finally averaged over the 15 annotators to yield the overall user-study results for each method.

## C.3    BASELINE ADAPTATIONS

We select four baselines: AnyV2V (Ku et al., 2024), VACE (Jiang et al., 2025), HunyuanCustom (Hu et al., 2025b) and Kling 1.6 Multimodal (Keling, 2025) for their robust stability, high accessiblity, and close relevance to our subject swapping objective.

Specifically, for AnyV2V, we edit the first frame as required using the latest image subject swapping method InsertAnything (Song et al., 2025), and utilize AnyV2V's default I2VGen-XL pipeline to obtain subsequent edited frames. For VACE and HunyuanCustom, we preprocess the mask sequences as specified by each method to ensure their best performance. For Kling 1.6 Multimodal, we use its official in-browser masking tool to define the source subject. Notably, this tool occasionally struggle with specific selection (e.g. garment inside the human region), limiting the number of valid annotations. We process as many instances from our benchmark as possible, yielding a total of 152 Kling results. Since these four methods all support text inputs but not solely rely on them, we provide the text prompts by briefly describing of the subject-reference pair.

PikaSwaps (Pika, 2025) is also a commercial model specializes in video subject swapping, but its API is consistently unavailable due to *high demand* during our whole evaluation period. If the API returns to normal operation, we will incorporate its results into benchmark at the earliest opportunity.

## D    ADDITIONAL EXPERIMENTS AND RESULTS

Figures 10-12 present extensive visual comparisons between DreamSwapV and the four baselines, spanning subject categories from *human* and *garment* to *small and large object*. Across all cases, our method delivers superior visual fidelity and stability, corroborating the state-of-the-art performance reported in the main paper.

## E    FURTHER DISCUSSION

### E.1    MORE APPLICATIONS

As noted in the main paper, our method can be extended to numerous related or downstream tasks, which we will analyze individually below.

**Image Subject Swapping.** Our framework is inherently image-compatible: a single image is treated as a frame=1 video. Figure 9a confirms that this simple adaptation yields high-fidelity image subject swapping.

**Video Inpainting or Addition.** As discussed in Section 3.1, slight changes in the input modality transform video subject swapping into video inpainting or video addition:

- *Inpainting.* If the reference image is omitted, the model automatically hallucinates plausible content inside the mask, restoring the damaged region (Figure 9b).

- *Addition.* Conversely, supplying an external mask sequence that covers empty space prompts the model to insert the reference object, effectively adding a moving subject to the clip (Figure 9c).

**Video Try-on.** When the target subject is constrained to garments, DreamSwapV acts as a video try-on system, as already illustrated in Figure 12. For more challenging scenarios like layered outfits and complex wrinkles, our model can serve as a strong prior and adapt after a brief quality tuning stage on a downstream try-on dataset, which we leave as future work.

**Hand-Object Interaction.** Limiting the subject to hand-held items turns the task into hand–object interaction (HOI) editing (Figure 11). Unlike template-based reference-to-video(R2V) pipelines such as AnchorCrafter (Xu et al., 2024c) or DreamActor-H1 (Wang et al., 2025), our approach directly swaps the product inside the original video—an avenue that remains unexplored. Same as extension to video try-on, with short, task-specific quality tuning, DreamSwapV could further bridge this gap and advance HOI video editing, a promising direction we also earmark for future exploration.

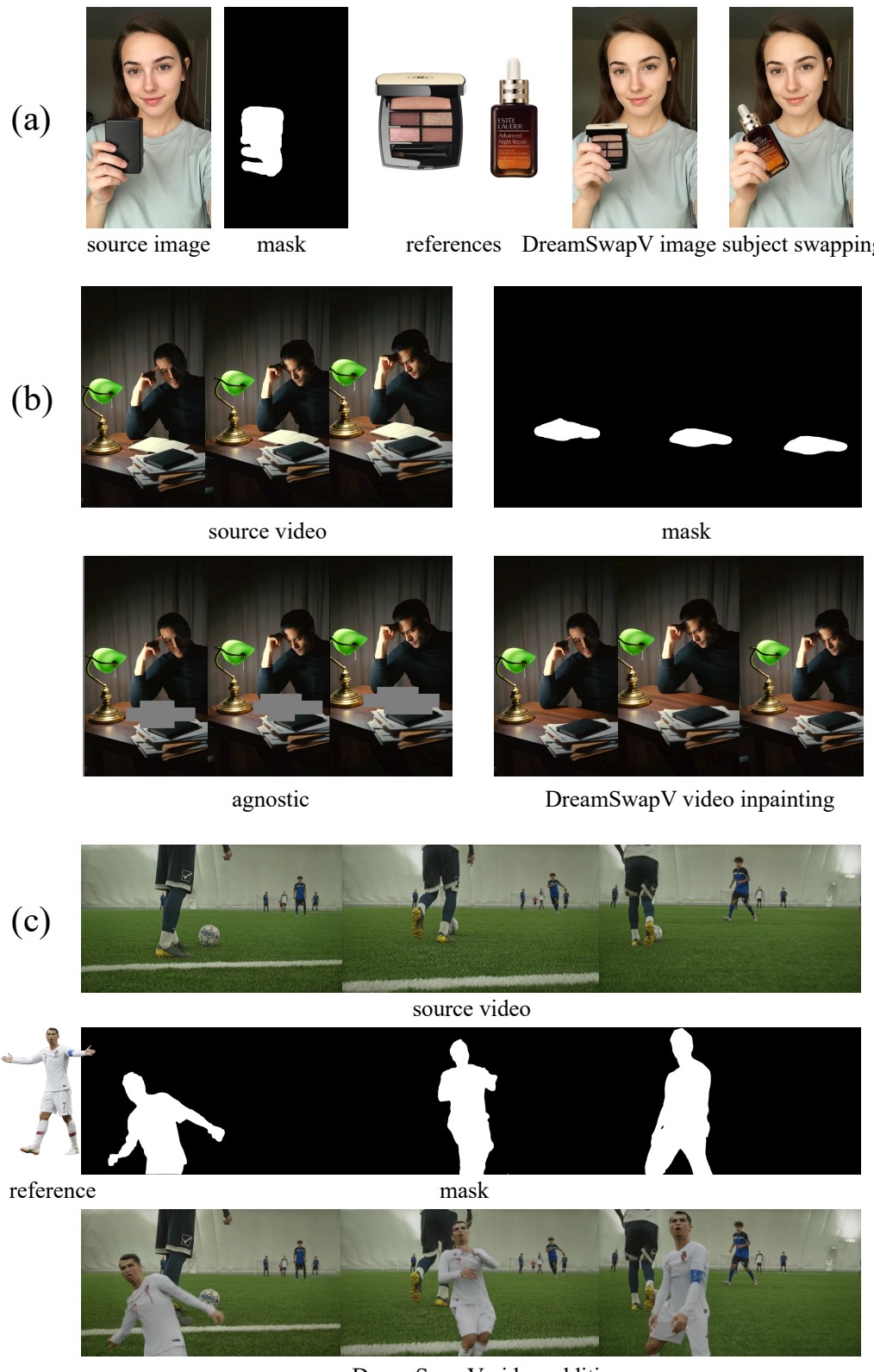

Figure 9: More applications of our DreamSwapV on (a) image subject swapping, (b) video inpainting, and (c) video addition, demonstrating the strong extensibility and transferability of our model to related and downstream tasks.

### E.2 SOCIAL IMPACT AND ETHICS STATEMENT

**Positive Impacts.** DreamSwapV broadens the reach of high-quality video subject swapping and related editing tasks by:

- **Lowering production barriers.** Content creators, educators and small studios can replace actors, props or garments without reshooting or green-screen setups, making professional video customization affordable and time-efficient.

- **Enabling new commercial and creative workflows.** The same pipeline supports video try-on, product insertion and damage inpainting, giving advertisers and e-commerce platforms rapid, personalized marketing assets while helping archivists restore legacy footage.

- **Research reusability.** Although DreamSwapV is initially fine-tuned from the Wan2.1-I2V-14B backbone (Wan et al., 2025), it remains model-independent: the framework is fully plug-and-play, allowing researchers to attach it to any DiT-based I2V model and potentially achieve even stronger subject-swapping performance.

**Risks and Challenges.** The same ease of use can facilitate malicious deepfakes, brand counterfeits, or non-consensual impersonations; attackers need only a reference photo to graft someone into misleading footage. Copyright conflicts may arise if trademarked characters or logos are swapped into commercial videos, and highly convincing forgeries could undermine public trust in authentic media.

**Ethical Concerns.** Responsible use requires securing permission from anyone depicted in reference images and respecting intellectual-property rights for all source material. Clear disclosure of edited content, along with basic provenance tools (e.g., watermarking or edit logs), helps viewers distinguish synthetic footage from genuine video.

### E.3 LARGE LANGUAGE MODEL USAGE

We use OpenAI's GPT-4o model (OpenAI, 2025) as a language assistant to improve the clarity and consistency of this manuscript. Its primary roles included:

- Language Polishing: Refining grammar, terminology, and sentence flow to better meet academic writing standards.

- Concise Rephrasing: Shortening verbose descriptions while preserving technical accuracy.

- LaTeX Formatting: Improving equation and table formatting in LaTeX for readability.

All outputs are reviewed and revised by the authors to ensure technical correctness. The model contributes no new scientific ideas or experimental results—its role is strictly limited to writing support.

### E.4 LIMITATIONS AND FUTURE WORK

Although fruitful, we need to admit that DreamSwapV still has several limitations. First, a single reference image conveys only limited information about the target subject; when the subject rotates or its backside is revealed, the swapping quality may degrade. Incorporating multi-view reference inputs—akin to AnchorCrafter(Xu et al., 2024c)—is a potential solution. Second, since rigid objects lack an explicit pose, fine-grained control remains challenging; introducing a spatial coordinate system (as in Orient Anything(Wang et al., 2024b)) to serve as the object's *pose* representation may yield more precise swapping. Third, the substantial training and inference costs of our full model may restrict its practicality for smaller-scale applications; we therefore plan to release a distilled version in the future for broader use. Finally, as discussed in Appendix E.1, adapting DreamSwapV to downstream tasks such as video try-on and hand–object interaction represents a promising research direction. In summary, while our current model has limitations, these potential enhancements open new pathways for future development. We eagerly anticipate the next explorations of DreamSwapV—let's stay tuned!

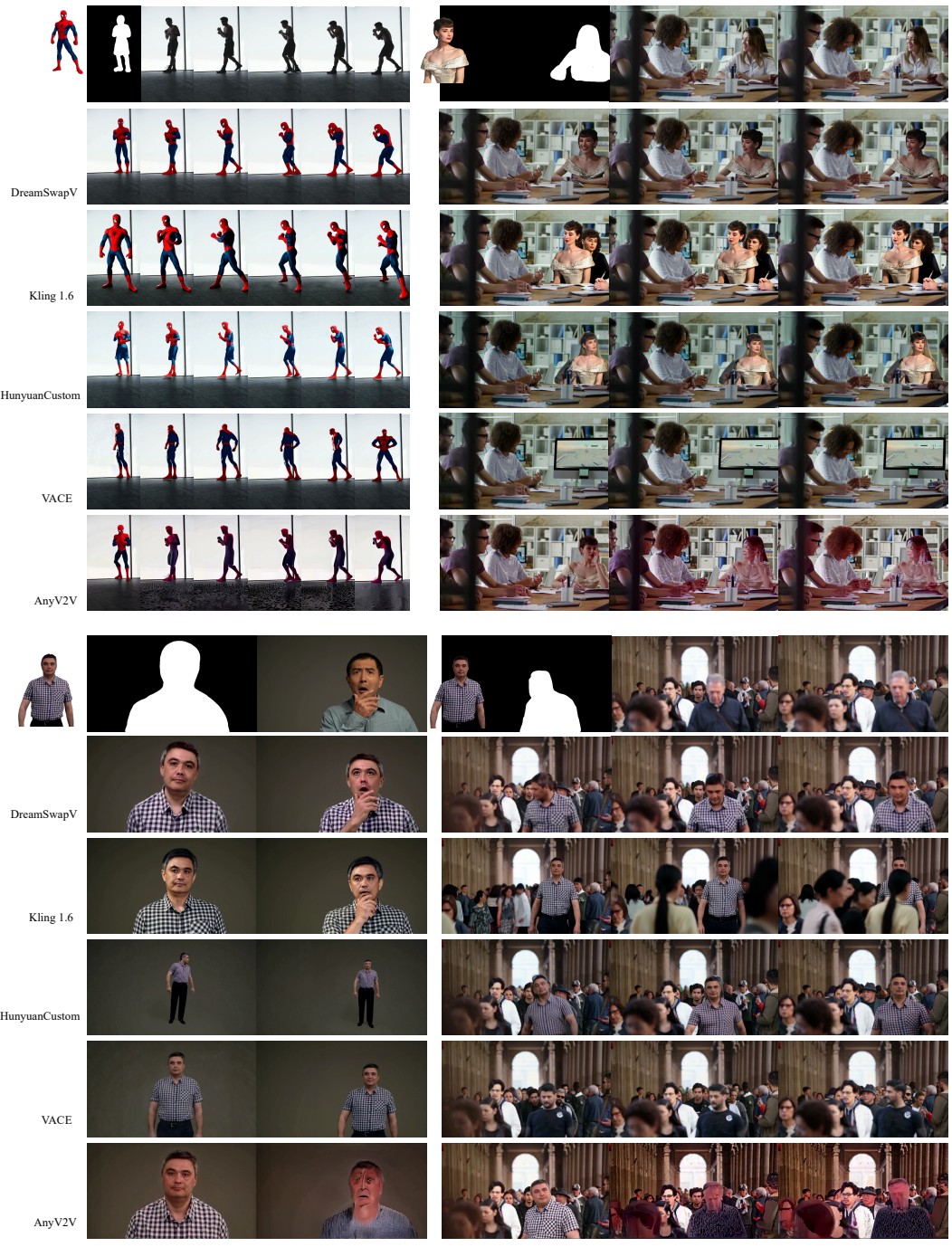

Figure 10: More qualitative comparisons between our DreamSwapV and four baselines in the *human* category (full body, half body and talking head). Please zoom in for details, and see the video demo for dynamic presentations.

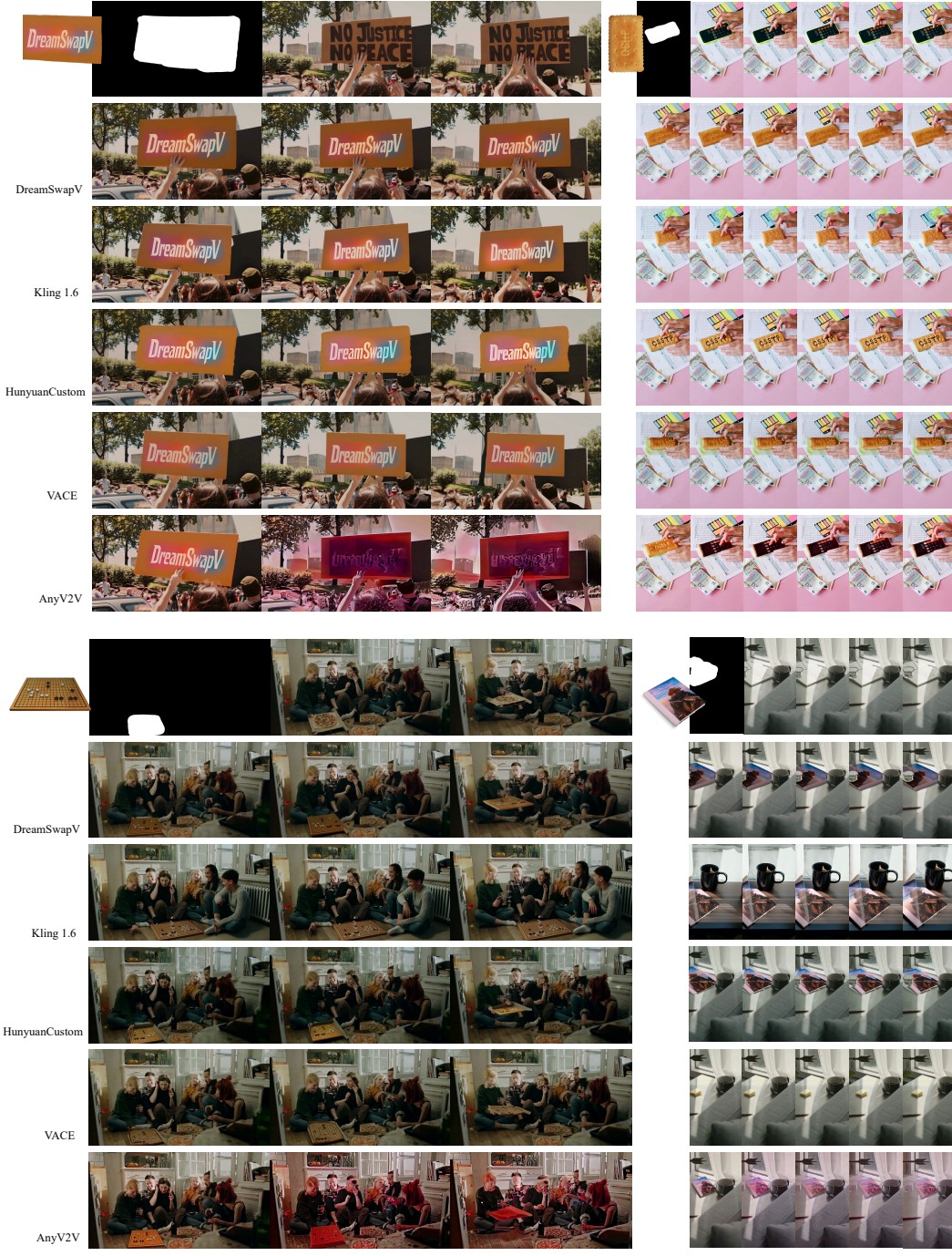

Figure 11: More qualitative comparisons between our DreamSwapV and four baselines in the *small object* category (handheld object and others). Please zoom in for details, and see the video demo for dynamic presentations.

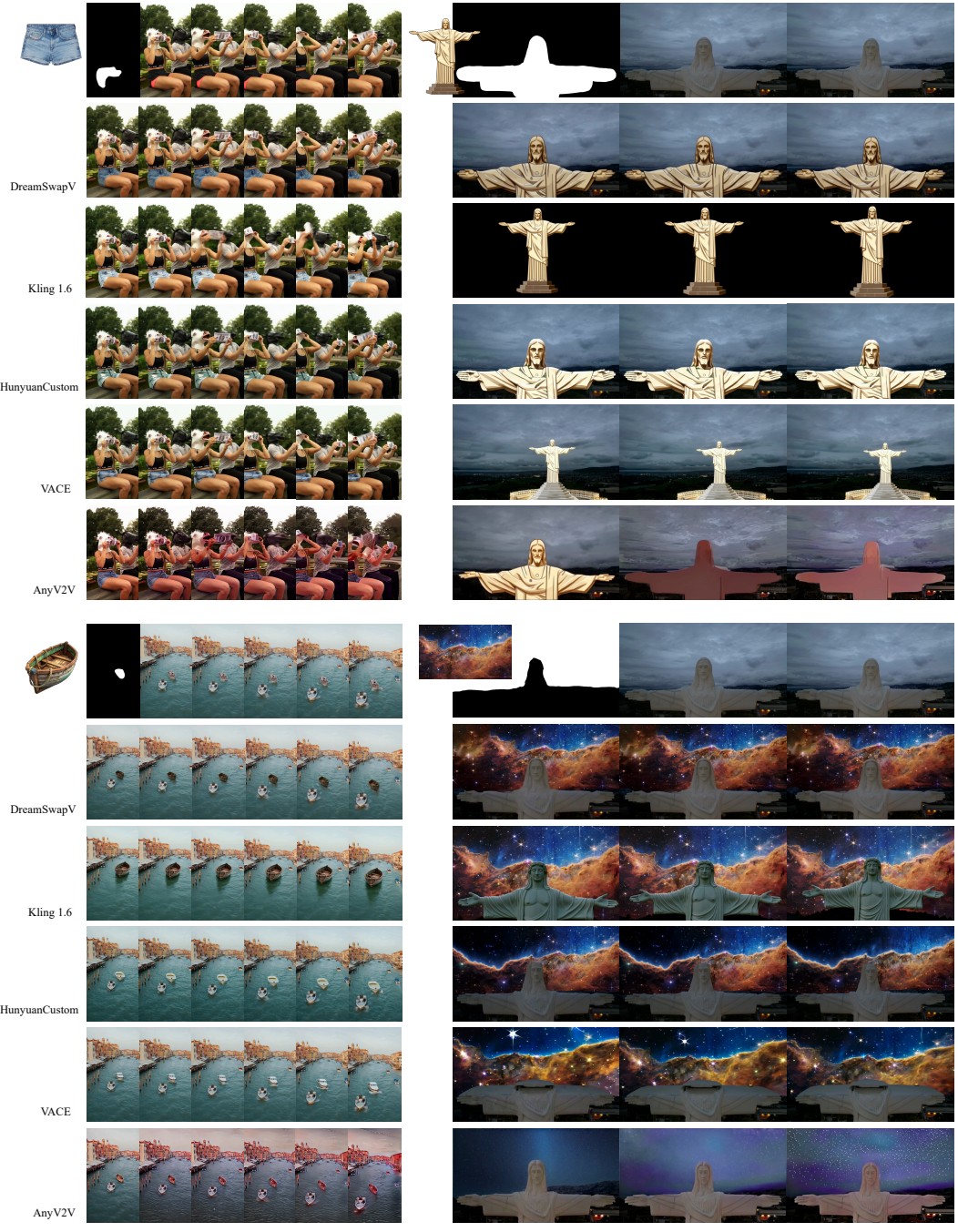

Figure 12: More qualitative comparisons between our DreamSwapV and four baselines in the *garment* and *large object* categories (vehicle, statue and sky). Please zoom in for details, and see the video demo for dynamic presentations.

