# OpenReview forum: "DreamSwapV: Mask-guided Subject Swapping for Any Customized Video Editing"
_ICLR.cc/2026/Conference — ICLR 2026 Poster_

### Official Review · Reviewer_bJk4 · 2025-10-25

**Soundness:** 3
**Presentation:** 3
**Contribution:** 2
**Rating:** 6
**Confidence:** 3

**Summary:**

This paper proposes an end-to-end framework based on mask guidance, which enables the replacement of any subject in a video using a specified mask and a reference object. Its key contributions are as follows:

- The proposal of an adaptive masking strategy to address practical issues.
- A dedicated conditional fusion module that integrates rich conditional information.
- The introduction of a new benchmark.

**Strengths:**

1. This paper has a clear logical flow, elaborating on its key contributions with a focus on two main aspects.
2. The adaptive masking strategy takes full account of various practical issues and achieves favorable results.
3. This paper demonstrates substantial research effort, solid argumentation, and has obtained excellent model performance.

**Weaknesses:**

1. There is still room for improvement and supplementation in the quantitative results of the experimental section of this paper.
2. The work of this paper mainly focuses on the targeted processing of data, with limited improvements to the model itself.

**Questions:**

1. In the Extra Shape Augmentation section, this paper mentions intentionally making the mask slightly larger than the target subject, which the authors consider a better approach. However, this approach is not supported by quantitative evidence in the subsequent experiments, and additional relevant experiments are expected to be supplemented.
2. This paper employs the newly proposed Benchmark for model evaluation in the experimental section. The evaluation results seem to be inconsistent with the quantitative results of Hunyuan Custom. Traditional quantitative comparison methods remain indispensable, and supplementary experiments in this regard are anticipated.

---

> ### Author Response · Authors · 2025-11-18
> **Initial Response to Reviewer bJk4**
>
> We would like to express our gratitude to Reviewer bJk4 for the thoughtful evaluation of our paper and offering helpful feedback.
>
> In this initial response, we focus on clarifying points that can be addressed immediately, and outline **[the necessary experiments]** that are underway to provide quantitative evidence, which will be reported in the next update.
>
> ### W1: Quantitative supplementation
> Based on the other reviewers' concerns, we will **supplement our quantitative experiments** in the following two aspects:
> - Additional metrics, such as *semantic consistency*, to complement the VBench metrics, as well as the *subject appearance* and *background preservation* metrics we proposed.
> - A quantitative comparison between the two training phases to demonstrate the effectiveness of our two-phase training strategy.
>
> **[These will be conducted using our proposed DreamSwapV-Benchmark and reported in the next update.]** If the reviewer has any further suggestions for quantitative improvement, we would be happy to receive them.
>
> ### W2: Model improvements
> Data is crucial for the final performance of our model, and we have indeed placed great emphasis on it by carefully constructing a two-stage dataset and designing two innovative modules: the *condition fusion module* and *adaptive mask strategy* to ensure effective utilization of the data. However, our work does not ***“mainly focus on”*** data.
>
> Regarding the discussion of reference information injection in Section 3.2—whether through channel concatenation, cross-attention, an additional ReferenceNet/adapter, or frame-wise concatenation (i.e. relying on self-attention)—this is **essentially about the model architecture**. Our final design (frame-wise concatenation with self-attention mask) is supported by both theoretical reasoning and empirical analysis, and happens to not require modifications to the underlying model architecture. This should not be interpreted as neglecting model improvements.
>
> Since this approach is **effective and minimally intrusive**, we see no reason not to adopt it; after all, modifying the model merely for the sake of modification is unnecessary. Moreover, leaving the architecture unchanged brings additional benefits: our entire framework can be readily transferred to other base models, such as subsequent versions of the WanX series (e.g., Wan 2.2) or alternatives like CogVideoX and HunyuanVideo. This contributes to **strong reproducibility, transferability, and community impact**, which is highlighted as important by Reviewer Wef7.
>
> ### Q1: Extra shape augmentation issues
> We would like to first clarify that making the mask slightly larger than the target subject is actually a **default setting**. Without any modification, the standard mask augmentation will naturally result in a mask that is slightly larger than the target subject. Animate Anyone 2 suggests an intentional adjustment, but we choose not to adopt this approach following the principle of minimal modification. We apologize for the misleading use of the word ***“intentionally”*** and will correct the wording.
>
> What we actually modify and aim to emphasize in this section is that maintaining the slightly larger mask can lead to hallucinations or confusion, which necessitates **the proposed extra shape augmentation**. We acknowledge that this section currently lacks some quantitative evidence, and we appreciate the reviewer's point on this matter. **[To avoid any misunderstandings, we will revise this section's focus in the final version and supplement our quantitative ablation study in the next update to demonstrate the effectiveness of the Extra Shape Augmentation.]**
>
> ### Q2: Inconsistent evaluation
> We need to make it clear that the evaluation of HunyuanCustom and ours are conducted on **entirely different tasks and domains**, and that's the reason why we need to construct our own benchmark. The results reported in Table 1 of HunyuanCustom are based on the single-subject video customization task, which is a [text+reference]-to-video setting. In contrast, our video subject swapping task corresponds to a reference-based video-to-video setting. These two tasks differ substantially in both inputs and objectives, and it is therefore natural that models exhibit different performance characteristics across them.
>
> Regarding *traditional quantitative comparison methods*, we consider VBench metrics to **fit into this category**, given that VBench is a well-established and widely adopted evaluation protocol in video quality assessment. It also incorporates many classical metrics, including DINO similarity, MAE, RAFT, aesthetic score, etc. If there are particular traditional metrics that the reviewer is especially concerned about, we are more than willing to provide their evaluation results on our DreamSwapV-Benchmark.
>
> We hope this initial response addresses your concerns and provides a deeper understanding of our method. We look forward to your feedback and further discussion!

---

> ### Author Response · Authors · 2025-11-21
> **Experimental Updates for Reviewer bJk4**
>
> Dear Reviewer bJk4,
>
> We hope you have already read our initial response, where we immediately clarified your questions and provided preliminary discussions. Over the past few days, we have conducted additional necessary experiments. Here, we report these new results to provide more substantial quantitative evidence.
>
> Your main concerns focus on:
> - Improvement and supplementation to the quantitative results of the experimental section
> - Quantitative ablation study on Extra Shape Augmentation
>
> We include all the experiments we have performed below (some conducted in response to concerns from other reviewers) so that you can have a thorough understanding of what we have done:
>
> | Method                | SC. | BC. | MS. | DD. | AQ. | VBench Avg. | RA. | BP. | **Semantic Consistency** | Total Avg. | **ID Preservation ↔** |
> |---------------------|:---:|:---:|:---:|:---:|:---:|:------------:|:---:|:---:|:---------------------:|:----------:|:-----------------:|
> | AnyV2V                | 90.03% | 91.35% | 98.60% | 47.90% | 51.79% | 75.93% | 34.70% | 42.71% | 51.00% | 63.51% | 14.48% |
> | VACE                  | 96.15% | *95.03%* | 99.29% | 27.54% | 56.95% | 74.99% | 39.66% | 47.46% | 66.93% | 66.16% | 48.66% |
> | HunyuanCustom         | 95.83% | 94.96% | 99.17% | 43.11% | **57.78%** | 78.17% | 41.33% | 48.14% | 63.65% | 68.00% | 35.23% |
> | Kling 1.6             | 95.36% | **96.57%** | **99.45%** | 50.33% | *57.26%* | 79.79% | 42.27% | 39.17% | *69.95%* | 68.80% | 43.10% |
> | DreamSwapV (Full)      | **96.41%** | 94.26% | *99.31%* | *55.69%* | 56.52% | **80.44%** | **45.22%** | **52.49%** | **72.01%** | **71.49%** | 40.71% |
> | **DreamSwapV (Phase 1)**   | *96.38%* | 94.04% | *99.31%* | *55.67%* | 55.37% | 80.15% | 43.52% | *52.41%* | 65.14% | 70.23% | 35.91% |
> | **w/o Extra Shape Aug.**  | 96.12% | 94.11% | 99.30% | 55.66% | 55.69% | *80.19%* | *43.61%* | 52.12% | 66.03% | *70.34%* | 30.28% |
>
> **Bold** indicates the highest score, and *italics* indicate the second-highest score. SC., BC., MS., DD., and AQ. denote the VBench metrics used in our main paper (*Subject Consistency, Background Consistency, Motion Smoothness, Dynamic Degree*, and *Aesthetic Quality*). RA. and BP. represent our proposed metrics: *Reference Appearance* and *Background Preservation*.
>
> The first five rows contain values preserved from the original main paper. We additionally provide two updated rows: **results from the Phase-1 model, and results from w/o Extra Shape Augmentation,**
> as extended ablation studies following your suggestion.
>
> Moreover, we introduce two supplementary metrics, ***Semantic Consistency*** and ***ID Preservation***, as suggested by Reviewer 5Z3c. *Semantic Consistency* is computed by captioning both the reference and the swapped subject (extracted using masks) via BLIP-2, then measuring caption similarity and normalizing the result. *ID Preservation* uses ArcFace embeddings of detected faces in the reference and swapped subject. It is similar to our *Reference Appearance* metric but focuses more on facial identity. For non-human subjects, this metric is skipped.
>
> Notably, *ID Preservation* is not a **“the higher, the better”** metric. Videos naturally require certain dynamics, and models that occasionally produce static or near-static content (e.g., VACE) tend to obtain artificially high ID Preservation scores—this is not a desirable behavior. For this reason, we do not include *ID Preservation* in the *Total Average* score (also not in the final main paper) and use the ↔ symbol to indicate that moderate values are preferred.
>
> In addition, we have updated the PDF and extended the manuscript to 10 pages. All modified or newly added content has been highlighted in ***steel blue*** for clear visibility. We strongly recommend the reviewer to refer to these highlighted sections to better understand the adjustments and improvements we have made in response to the review comments during the rebuttal stage.
>
> We sincerely hope that the additional quantitative experiments, extended discussions, and updated manuscript collectively address your concerns. If you have any further questions or suggestions, please feel free to let us know — we would be more than happy to provide additional clarification or conduct further discussion.

---

### Official Review · Reviewer_Wef7 · 2025-10-26

**Soundness:** 3
**Presentation:** 3
**Contribution:** 3
**Rating:** 6
**Confidence:** 3

**Summary:**

This paper presents DreamSwapV, which is a mask-guided and subject-agnostic framework for video subject swapping. It allows users to replace any subject in any video using a mask and a reference image. The method treats swapping as a video inpainting task, enabling seamless blending between the new subject and the original scene. A condition fusion module integrates mask, pose, and 3D-hand inputs, while an adaptive mask strategy handles subjects of different sizes. The model is trained in two phases for better generalization. Experiments on a new DreamSwapV-Benchmark show that it achieves higher visual quality and consistency than existing methods.

**Strengths:**

1. The proposal of the condition fusion module and adaptive mask strategy enables fine-grained control, resulting better subject-context interaction together with high-quality visual improvements.
2. The paper introduces a new benchmark DreamSwapV-Benchmark in customized image editing domain and the proposed method shows improvements over previous baselines with both quantitative metrics and user studies.
3. The model can be extended beyond subject swapping to related tasks like video inpainting, addition, and try-on, which may show good generalization potential.

**Weaknesses:**

1. The method relies on many detailed design choices and a two-phase training process across multiple datasets. While these improve performance, they make the system complicated and harder to reproduce.
2. Some baselines, like AnyV2V, are training-free or designed for broader editing rather than subject swapping, which makes the comparison less fair.

**Questions:**

1. Can you provide some failure case analysis for your method?
2. The two-phase training scheme and dataset mixture seem central to performance. Could the authors clarify how much improvement the second phase brings quantitatively, and whether similar performance could be achieved with a single large-scale fine-tuning?

---

> ### Author Response · Authors · 2025-11-18
> **Initial Response to Reviewer Wef7**
>
> We sincerely appreciate Reviewer Wef7 for the careful reading of our work and providing insightful feedback.
>
> In this initial response, we first clarify the points that can be directly addressed without additional experiments, aiming to facilitate an efficient and in-depth discussion. We also outline **[the necessary experiments]**, which will provide empirical evidence for the remaining questions and will be included in the next update.
>
> ### W1: Reproducibility
> To achieve **state-of-the-art and stable performance**, we carefully implement several detailed design choices, which we view as one of the contributions of our method. However, our system is not ***"complicated"*** despite these detailed designs—each component is **intuitive, self-contained, and easy to implement**. Importantly, we do not modify or depend on the architecture of the underlying model (Wan 2.1 I2V). This means our entire framework can be readily transferred to other base models, such as subsequent versions of the WanX series (e.g., Wan 2.2) or potential alternatives like CogVideoX and HunyuanVideo. Other video-related tasks (such as reference-to-video, video try-on, etc.) may also benefit from our design principles, reusing them in their own methods. All of the above contribute to **strong reproducibility, transferability, and community impact**. After acceptance, we will further release the inference code and model checkpoints to facilitate reproducibility and future research in the community.
>
> ### W2: Less fair comparison
> We acknowledge that some baselines used for comparison are training-free or designed for broader editing tasks. However, since no dedicated video subject swapping models currently exist, we **have no choice but to compare** these methods due to their strong editing capabilities and robust performance. We have already chosen the current state-of-the-art editing methods and, even so, identify their limitations, which underscores the **novelty** of our work and the **gap it fills** in this field.
>
> Notably, we do not intend to criticize these methods; on the contrary, we deeply appreciate and respect their contributions to training-free and broader editing tasks. They continue to perform well in their respective target domains, but are outperformed by our method on the video subject swapping task, which is our **primary target**. Overall, our comparison is not intended to be unfair (unlike deliberately selecting weaker baselines or tweaking hyperparameters).
>
> ### Q1: Failure cases
> As partially discussed in Section 3.4, our failure cases primarily arise when **handling cross-domain swapping modes**, which are rare in our training data. For instance, inserting a half-body human into a full-body mask causes the model to fabricate details due to the lack of relevant information. Meanwhile, in cases such as animal → human or object → character, the poses of animals or objects differ significantly from those of humans, further challenging our pose-conditioned model. We have attempted to introduce such cases during training to help the model adapt to these more complicated swapping modes, but observed hallucinations and detail loss as a result of the increased learning difficulty. Therefore, to prioritize high-quality in-domain swapping, we did not include these cross-domain cases in training, and cross-domain swapping remains an unsolved challenge as a **trade-off**. We leave the task of cross-domain swapping to future mask-free methods.
>
> ### Q2: Comparison between phases
> We apologize for not including the comparison between the two training phases. This should (and will) be included in the main ablation study as evidence of the effectiveness of our two-phase training strategy. We appreciate the reviewer for highlighting this point. **[Fortunately, we saved the checkpoint from the first training phase and will evaluate it using the proposed DreamSwapV-Benchmark. The results will be reported in the next update.]**
>
> Regarding whether similar performance could be achieved with a single large-scale fine-tuning, we discuss the necessity of the second-stage quality-tuning data in Section 3.4. This data is carefully curated with higher-quality videos, and unlike the pre-training set, the references are not directly cropped from the videos, preventing the model from learning **a trivial copy-paste behavior**. Due to the large scale gap, merging the datasets would **dilute** the important patterns in the high-quality data. Our two-stage strategy first adapts the image-to-video model to video subject swapping using large-scale pre-training data with only self-attention trained, then refines detailed swapping patterns with small-scale, high-quality data via full fine-tuning. Naively combining them would confuse these stage-specific objectives and lead to inefficient learning.
>
> We hope this initial response addresses your concerns and provides a deeper understanding of our method. We look forward to your feedback and further discussion!

---

> ### Author Response · Authors · 2025-11-21
> **Experimental Updates for Reviewer Wef7**
>
> Dear Reviewer Wef7,
>
> We hope you have already read our initial response, where we immediately clarified your questions and provided preliminary discussions. Over the past few days, we have conducted additional necessary experiments. Here, we report these new results to provide more substantial quantitative evidence.
>
> Your main concern focuses on:
> - Quantitative comparison between two training phases
>
> We include all the experiments we have performed below (some conducted in response to concerns from other reviewers) so that you can have a thorough understanding of what we have done:
>
> | Method                | SC. | BC. | MS. | DD. | AQ. | VBench Avg. | RA. | BP. | **Semantic Consistency** | Total Avg. | **ID Preservation ↔** |
> |---------------------|:---:|:---:|:---:|:---:|:---:|:------------:|:---:|:---:|:---------------------:|:----------:|:-----------------:|
> | AnyV2V                | 90.03% | 91.35% | 98.60% | 47.90% | 51.79% | 75.93% | 34.70% | 42.71% | 51.00% | 63.51% | 14.48% |
> | VACE                  | 96.15% | *95.03%* | 99.29% | 27.54% | 56.95% | 74.99% | 39.66% | 47.46% | 66.93% | 66.16% | 48.66% |
> | HunyuanCustom         | 95.83% | 94.96% | 99.17% | 43.11% | **57.78%** | 78.17% | 41.33% | 48.14% | 63.65% | 68.00% | 35.23% |
> | Kling 1.6             | 95.36% | **96.57%** | **99.45%** | 50.33% | *57.26%* | 79.79% | 42.27% | 39.17% | *69.95%* | 68.80% | 43.10% |
> | DreamSwapV (Full)      | **96.41%** | 94.26% | *99.31%* | *55.69%* | 56.52% | **80.44%** | **45.22%** | **52.49%** | **72.01%** | **71.49%** | 40.71% |
> | **DreamSwapV (Phase 1)**   | *96.38%* | 94.04% | *99.31%* | *55.67%* | 55.37% | 80.15% | 43.52% | *52.41%* | 65.14% | 70.23% | 35.91% |
> | **w/o Extra Shape Aug.**  | 96.12% | 94.11% | 99.30% | 55.66% | 55.69% | *80.19%* | *43.61%* | 52.12% | 66.03% | *70.34%* | 30.28% |
>
> **Bold** indicates the highest score, and *italics* indicate the second-highest score. SC., BC., MS., DD., and AQ. denote the VBench metrics used in our main paper (*Subject Consistency, Background Consistency, Motion Smoothness, Dynamic Degree*, and *Aesthetic Quality*). RA. and BP. represent our proposed metrics: *Reference Appearance* and *Background Preservation*.
>
> The first five rows contain values preserved from the original main paper. We additionally provide two updated rows: **results from the Phase-1 model, and results from w/o Extra Shape Augmentation,**
> as extended ablation studies.
>
> Moreover, we introduce two supplementary metrics, ***Semantic Consistency*** and ***ID Preservation***, as suggested by Reviewer 5Z3c. *Semantic Consistency* is computed by captioning both the reference and the swapped subject (extracted using masks) via BLIP-2, then measuring caption similarity and normalizing the result. *ID Preservation* uses ArcFace embeddings of detected faces in the reference and swapped subject. It is similar to our *Reference Appearance* metric but focuses more on facial identity. For non-human subjects, this metric is skipped.
>
> Notably, *ID Preservation* is not a **“the higher, the better”** metric. Videos naturally require certain dynamics, and models that occasionally produce static or near-static content (e.g., VACE) tend to obtain artificially high ID Preservation scores—this is not a desirable behavior. For this reason, we do not include *ID Preservation* in the *Total Average* score (also not in the final main paper) and use the ↔ symbol to indicate that moderate values are preferred.
>
> In addition, we have updated the PDF and extended the manuscript to 10 pages. All modified or newly added content has been highlighted in ***steel blue*** for clear visibility. We strongly recommend the reviewer to refer to these highlighted sections to better understand the adjustments and improvements we have made in response to the review comments during the rebuttal stage.
>
> We sincerely hope that the additional quantitative experiments, extended discussions, and updated manuscript collectively address your concerns. If you have any further questions or suggestions, please feel free to let us know — we would be more than happy to provide additional clarification or conduct further discussion.

---

> > ### Comment · Reviewer_Wef7 · 2025-11-26
> >
> > Thanks for the reply from the authors. I will increase my score correspondingly.

---

> > > ### Author Response · Authors · 2025-11-27
> > >
> > > Thank you very much for your continued support and for the increased score. We truly appreciate your insightful feedback throughout the review process, which has helped us further strengthen this work. If you have any further suggestions, we would warmly welcome them!

---

### Official Review · Reviewer_5Z3c · 2025-10-30

**Soundness:** 3
**Presentation:** 3
**Contribution:** 2
**Rating:** 6
**Confidence:** 4

**Summary:**

The paper presents DreamSwapV, a mask-guided, subject-agnostic framework for end-to-end video subject swapping. The method reformulates the swapping process as a video inpainting task, enabling more seamless integration between the inserted subject and the background. The proposed system combines a condition fusion module that integrates multiple signals and an adaptive mask strategy that accommodates subjects of varying scales.

**Strengths:**

* The combination of a multi-condition fusion module and adaptive mask strategy is intuitive yet effective, addressing common artifacts, e.g., boundary leakage and poor subject-context blending.

* The proposed DreamSwapV-Benchmark is a valuable addition to the field, with well-defined metrics and an attempt at quantitative evaluation for a task lacking standard benchmarks.

* The paper is clearly written and easy to follow, with well-organized figures and methodological explanations.

**Weaknesses:**

* While the new benchmark is appreciated, the evaluation relies heavily on VBench-style metrics that may not capture identity preservation or semantic consistency robustly. A few qualitative examples are shown, but it remains unclear how the model generalizes to truly out-of-domain subjects or complex dynamic interactions.
* Some ablations, e.g., condition fusion variants, mask augmentation parameters, are only qualitatively discussed. Quantitative ablations would make the argument stronger.
* The method is heavily trained and evaluated on HumanVID-derived data, raising concerns about its generalization to more diverse subjects.

**Questions:**

My key questions are related to the weaknesses mentioned above; besides those, I still have a few minor questions.
* How does the model perform on truly cross-domain swapping (e.g., animal → human or object → character), and what failure modes are observed?
* How does the method behave under mask noise or misalignment, such as slightly inaccurate user-provided masks — does performance degrade sharply or remain stable?

---

> ### Author Response · Authors · 2025-11-18
> **Initial Response to Reviewer 5Z3c**
>
> We are grateful to Reviewer 5Z3c for taking the time to review our paper and for offering thoughtful and insightful remarks.
>
> In this initial response, we focus on clarifying issues that can be addressed immediately to support an efficient discussion process. We also outline **[the necessary experiments]** under progress, which will be reported in the next update.
>
> ### W1: Other metrics and model generalization
>
> We choose VBench for its well-established protocol, primarily to assess low-level video quality aspects (e.g., dynamics, motion smoothness, aesthetics), which remain important in comprehensive video evaluation. However, we would like to clarify that our evaluation does not ***“heavily rely on”*** these metrics. As discussed in the paper, we have realized that VBench is not fully suited for the subject swapping task, so we additionally include *subject appearance* and *background preservation*. Among them, *subject appearance* can measure the reviewer’s concern regarding *identity preservation*, as it computes the DINO similarity between the swapped region subject and the reference. **[However, we acknowledge the value of incorporating additional metrics, and we will include *semantic consistency* along with ArcFace similarity (for finer *identity preservation* evaluation) in the next update.]**
>
> Regarding generalization to out-of-domain subjects and complex dynamic interactions, we have provided **additional qualitative examples across diverse categories** (*human, garment, handheld and large object*) in both the appendix after the main paper and the supplementary video demo. Among them, many subjects such as *cartoon characters*, *vehicles* and *buildings* rarely appear in our real-world, human-centric HumanVID-derived dataset, demonstrating some generalization to out-of-domain subjects. For complex dynamic interactions, we recommend checking the handheld swapping examples—*Pocky* in the main paper, *Parade sign* and *Biscuit* in the appendix. We are **the only method** that successfully handles fine-grained hand–object interactions, thanks to our pose condition fusion and adaptive mask strategy. Based on the reviewer's concern, we will move some of these qualitative results into the main paper to provide a more comprehensive presentation. We also acknowledge that extremely out-of-domain cases remain challenging for the model; please refer to **Q1** for a detailed discussion.
>
> ### W2: Quantitative ablation
> We have provided **a quantitative ablation study** on condition fusion variants and adaptive grid sizing in Table 3 of the appendix (not in the main paper limited by space), evaluated on our DreamSwapV-Benchmark, which will be moved into the main paper in the final version.
>
> ### W3: Generalization to diverse subjects
> Please refer to the discussion on out-of-domain subjects in **W1**.
>
> ### Q1: Cross-domain swapping and failure modes
> As partially discussed in Section 3.4, certain cross-domain swapping modes (e.g., inserting a half-body human into a full-body mask) **pose challenges** for our model. We attempted random cropping to expose the model to some of these corner cases, but the increased learning difficulty led to a noticeable performance degradation (hallucinations and detail loss as failure modes). To prioritize high-quality in-domain swapping, we did not include such cross-domain attempts during training, and thus cross-domain swapping remains unsolved as a **trade-off**. For cases such as animal → human or object → character, the poses of animals or objects differ significantly from humans, further increasing the difficulty for our pose-conditioned approach. We leave the task of cross-domain swapping to future mask-free work.
>
> ### Q2: Mask noise or inaccuracies
> Since our model is strictly trained in a mask-guided manner, the accuracy of the mask is **crucial** for final performance. Any regions not covered by the mask will be strictly preserved and referenced. If the mask contains certain inaccuracies that leak parts of the source subject, the model may reference these regions during swapping, potentially reconstructing the source subject rather than injecting the external reference. Even if the swapping succeeds, the presence of unmasked source content can impair visual quality.
>
> To mitigate this issue, besides providing a more precise mask sequence, we encourage users to employ our **bounding-box inference mode**, which expands the mask to the bounding box of the source subject to ensure full coverage. Since our model has been exposed to this mode during training, it can reliably handle bounding-box-based swapping. Although the loss of fine-grained mask-edge control may introduce slight performance degradation, this mode can **effectively reduce swapping failures** caused by mask inaccuracies.
>
> We hope this initial response addresses your concerns and provides a deeper understanding of our method. We look forward to your feedback and further discussion!

---

> ### Author Response · Authors · 2025-11-21
> **Experimental Updates for Reviewer 5Z3c**
>
> Dear Reviewer 5Z3c,
>
> We hope you have already read our initial response, where we immediately clarified your questions and provided preliminary discussions. Over the past few days, we have conducted additional necessary experiments. Here, we report these new results to provide more substantial quantitative evidence.
>
> Your main concern focuses on:
> - Complementary metrics like *ID Preservation* and *Semantic Consistency* in support of VBench metrics
>
> We include all the experiments we have performed below (some conducted in response to concerns from other reviewers) so that you can have a thorough understanding of what we have done:
>
> | Method                | SC. | BC. | MS. | DD. | AQ. | VBench Avg. | RA. | BP. | **Semantic Consistency** | Total Avg. | **ID Preservation ↔** |
> |---------------------|:---:|:---:|:---:|:---:|:---:|:------------:|:---:|:---:|:---------------------:|:----------:|:-----------------:|
> | AnyV2V                | 90.03% | 91.35% | 98.60% | 47.90% | 51.79% | 75.93% | 34.70% | 42.71% | 51.00% | 63.51% | 14.48% |
> | VACE                  | 96.15% | *95.03%* | 99.29% | 27.54% | 56.95% | 74.99% | 39.66% | 47.46% | 66.93% | 66.16% | 48.66% |
> | HunyuanCustom         | 95.83% | 94.96% | 99.17% | 43.11% | **57.78%** | 78.17% | 41.33% | 48.14% | 63.65% | 68.00% | 35.23% |
> | Kling 1.6             | 95.36% | **96.57%** | **99.45%** | 50.33% | *57.26%* | 79.79% | 42.27% | 39.17% | *69.95%* | 68.80% | 43.10% |
> | DreamSwapV (Full)      | **96.41%** | 94.26% | *99.31%* | *55.69%* | 56.52% | **80.44%** | **45.22%** | **52.49%** | **72.01%** | **71.49%** | 40.71% |
> | **DreamSwapV (Phase 1)**   | *96.38%* | 94.04% | *99.31%* | *55.67%* | 55.37% | 80.15% | 43.52% | *52.41%* | 65.14% | 70.23% | 35.91% |
> | **w/o Extra Shape Aug.**  | 96.12% | 94.11% | 99.30% | 55.66% | 55.69% | *80.19%* | *43.61%* | 52.12% | 66.03% | *70.34%* | 30.28% |
>
> **Bold** indicates the highest score, and *italics* indicate the second-highest score. SC., BC., MS., DD., and AQ. denote the VBench metrics used in our main paper (*Subject Consistency, Background Consistency, Motion Smoothness, Dynamic Degree*, and *Aesthetic Quality*). RA. and BP. represent our proposed metrics: *Reference Appearance* and *Background Preservation*.
>
> The first five rows contain values preserved from the original main paper. We additionally provide two updated rows: **results from the Phase-1 model, and results from w/o Extra Shape Augmentation,**
> as extended ablation studies suggested by other reviewers.
>
> Moreover, we introduce two supplementary metrics, ***Semantic Consistency*** and ***ID Preservation***, following your suggestion. *Semantic Consistency* is computed by captioning both the reference and the swapped subject (extracted using masks) via BLIP-2, then measuring caption similarity and normalizing the result. *ID Preservation* uses ArcFace embeddings of detected faces in the reference and swapped subject. It is similar to our *Reference Appearance* metric but focuses more on facial identity. For non-human subjects, this metric is skipped.
>
> Notably, *ID Preservation* is not a **“the higher, the better”** metric. Videos naturally require certain dynamics, and models that occasionally produce static or near-static content (e.g., VACE) tend to obtain artificially high ID Preservation scores—this is not a desirable behavior. For this reason, we do not include *ID Preservation* in the *Total Average* score (also not in the final main paper) and use the ↔ symbol to indicate that moderate values are preferred.
>
> In addition, we have updated the PDF and extended the manuscript to 10 pages. All modified or newly added content has been highlighted in ***steel blue*** for clear visibility. We strongly recommend the reviewer to refer to these highlighted sections to better understand the adjustments and improvements we have made in response to the review comments during the rebuttal stage.
>
> We sincerely hope that the additional quantitative experiments, extended discussions, and updated manuscript collectively address your concerns. If you have any further questions or suggestions, please feel free to let us know — we would be more than happy to provide additional clarification or conduct further discussion.

---

### Official Review · Reviewer_pJC7 · 2025-11-01

**Soundness:** 2
**Presentation:** 2
**Contribution:** 2
**Rating:** 4
**Confidence:** 5

**Summary:**

This paper proposed DreamSwapV, an mask-guided framework for video subject swapping that allows users to swap any subject in any video using a mask and reference image. It introduced a multi-condition fusion module and an adaptive mask strategy to handle subjects of varying scales and attributes. DreamSwapV leverages an elaborate two-phase training scheme on a carefully curated dataset and introduced the DreamSwapV-Benchmark, and achieving best performance on the benchmark.

**Strengths:**

S1) The method is well ablated and justified the design choices that yields finer details and more robust subject-context integration.

S2) Adaptive mask makes a good design by dynamically adjusting the grid size based on the subject's scale and augmenting mask boundaries with geometric shapes.

S3) Its great to see the discussion on handling long videos, as most related works can only edit videos on a certain length / training length.

**Weaknesses:**

W1) In Table 1, the automatic metrics "Video Quality & Video Consistency" show AnyV2V with high scores on most quantitative indicators (subject consistency, background consistency, motion smoothness), nearly on par with other leading methods. However, the user study metric (human-rated reference detail, subject interaction, and visual fidelity) shows AnyV2V achieving nearly zero. This shows the proposed metrics are insufficiently sensitive to qualitative breakdowns. It seems only the reference appearance and background preservation metrics are making sense here. The authors might consider replacing VBench metrics with alternatives.

W2) Since the method is guided on the masks, the quality of the mask directly affects the final swapping results. Although detection models like TrackingSAM can handle the mask, but any errors could happen in fast motions and propagate to the downstream swapping result. Any remedy to make sure the mask is correctly handled?

**Questions:**

Q1) Since phase 1 is mainly trained on human (HumanVID), and phase 2 is trained on small subjects (Subject200K etc), this raise an interesting question. How much gain did the model get after each stage? Comparing the model’s performance after each training phase would give more insights on how effective this method is.

---

> ### Author Response · Authors · 2025-11-13
> **Initial Response to Reviewer pJC7**
>
> We sincerely thank Reviewer pJC7 for the time and effort devoted to carefully reading our paper and for providing valuable and insightful comments.
>
> In this initial response, we would like to first address the questions that can be clarified immediately to facilitate a timely and in-depth discussion. We also outline **[the necessary experiments]** that are currently in progress to more clearly address questions requiring empirical evidence, which will be reported in the next update.
>
> ### W1: VBench issues and alternatives
> VBench is a well-established benchmark in video quality assessment, so we use it to evaluate **the fundamental aspects** of generated videos (e.g., dynamic degree and aesthetics). However, as correctly pointed out by the reviewer, VBench metrics are not sufficiently sensitive to the qualitative aspects specific to the video subject swapping task. This limitation is precisely why we additionally propose **two task-specific metrics** — *reference appearance* and *background preservation* — which are not part of VBench.
>
> However, VBench **remains important** for evaluating low-level video quality. For example, VACE occasionally produces almost static videos — leading to a much lower score in *dynamic degree* compared to AnyV2V, which is reasonable even though AnyV2V may fail in video subject swapping. Moreover, the differences among methods on several VBench metrics are **generally small**; as reported in the original VBench paper, metrics such as *motion smoothness* and *temporal flickering* often vary by only a few percentage points between the best and worst-performing models. Thus, the ***“nearly on par with”*** observation still reflects a meaningful gap. **[To further supplement VBench (with alternatives), we will introduce additional metrics such as semantic consistency following Reviewer 5Z3c’s suggestion, which will be reported in the next update.]**
>
> ### W2: Mask errors remedy
> We agree that hard cases (e.g., fast-moving or heavily occluded objects) may cause TrackingSAM to produce inaccurate masks. To mitigate this, during inference time we assume that consecutive masks should **share some spatial overlap**. If two adjacent masks show little overlap (< 5% in practice), we treat it as a potential prediction error and take their **minimal enclosing region** to ensure the target subject remains covered. With abundant information from other frames, this strategy gives the model **flexibility** to distinguish the swapped object from the original background under other conditions (e.g., pose). We thank the reviewer for highlighting this issue and will include a discussion of this mechanism in the revised version. In such cases, users are also encouraged to provide their precise mask sequence for optimal results.
>
> ### Q1: Comparison between phases
> We apologize for the oversight of the comparison between the two training phases. This should (and will) be included in the main ablation study as evidence of the effectiveness of our two-phase training strategy, and we thank the reviewer for pointing this out. **[Fortunately, we saved the checkpoint of the first training phase and will evaluate it on the proposed DreamSwapV-Benchmark, with the results to be reported in the next update.]**
>
> We hope this initial response addresses your concerns and provides a deeper understanding of our method. We look forward to your feedback and further discussion!

---

> ### Author Response · Authors · 2025-11-21
> **Experimental Updates for Reviewer pJC7**
>
> Dear Reviewer pJC7,
>
> We hope you have already read our initial response, where we immediately clarified your questions and provided preliminary discussions. Over the past few days, we have conducted additional necessary experiments. Here, we report these new results to provide more substantial quantitative evidence.
>
> Your main concerns focus on:
> - Quantitative comparison between two training phases
> - Complementary metrics beyond VBench
>
> We include all the experiments we have performed below (some conducted in response to concerns from other reviewers) so that you can have a thorough understanding of what we have done:
>
> | Method                | SC. | BC. | MS. | DD. | AQ. | VBench Avg. | RA. | BP. | **Semantic Consistency** | Total Avg. | **ID Preservation ↔** |
> |---------------------|:---:|:---:|:---:|:---:|:---:|:------------:|:---:|:---:|:---------------------:|:----------:|:-----------------:|
> | AnyV2V                | 90.03% | 91.35% | 98.60% | 47.90% | 51.79% | 75.93% | 34.70% | 42.71% | 51.00% | 63.51% | 14.48% |
> | VACE                  | 96.15% | *95.03%* | 99.29% | 27.54% | 56.95% | 74.99% | 39.66% | 47.46% | 66.93% | 66.16% | 48.66% |
> | HunyuanCustom         | 95.83% | 94.96% | 99.17% | 43.11% | **57.78%** | 78.17% | 41.33% | 48.14% | 63.65% | 68.00% | 35.23% |
> | Kling 1.6             | 95.36% | **96.57%** | **99.45%** | 50.33% | *57.26%* | 79.79% | 42.27% | 39.17% | *69.95%* | 68.80% | 43.10% |
> | DreamSwapV (Full)      | **96.41%** | 94.26% | *99.31%* | *55.69%* | 56.52% | **80.44%** | **45.22%** | **52.49%** | **72.01%** | **71.49%** | 40.71% |
> | **DreamSwapV (Phase-1)**   | *96.38%* | 94.04% | *99.31%* | *55.67%* | 55.37% | 80.15% | 43.52% | *52.41%* | 65.14% | 70.23% | 35.91% |
> | **w/o Extra Shape Aug.**  | 96.12% | 94.11% | 99.30% | 55.66% | 55.69% | *80.19%* | *43.61%* | 52.12% | 66.03% | *70.34%* | 30.28% |
>
> **Bold** indicates the highest score, and *italics* indicate the second-highest score. SC., BC., MS., DD., and AQ. denote the VBench metrics used in our main paper (*Subject Consistency, Background Consistency, Motion Smoothness, Dynamic Degree*, and *Aesthetic Quality*). RA. and BP. represent our proposed metrics: *Reference Appearance* and *Background Preservation*.
>
> The first five rows contain values preserved from the original main paper. We additionally provide two updated rows: **results from the Phase-1 model, and results from w/o Extra Shape Augmentation,**
> as extended ablation studies.
>
> Moreover, we introduce two supplementary metrics, ***Semantic Consistency*** and ***ID Preservation***, as suggested by Reviewer 5Z3c. *Semantic Consistency* is computed by captioning both the reference and the swapped subject (extracted using masks) via BLIP-2, then measuring caption similarity and normalizing the result. *ID Preservation* uses ArcFace embeddings of detected faces in the reference and swapped subject. It is similar to our *Reference Appearance* metric but focuses more on facial identity. For non-human subjects, this metric is skipped.
>
> Notably, *ID Preservation* is not a **“the higher, the better”** metric. Videos naturally require certain dynamics, and models that occasionally produce static or near-static content (e.g., VACE) tend to obtain artificially high ID Preservation scores—this is not a desirable behavior. For this reason, we do not include *ID Preservation* in the *Total Average* score (also not in the final main paper) and use the ↔ symbol to indicate that moderate values are preferred.
>
> In addition, we have updated the PDF and extended the manuscript to 10 pages. All modified or newly added content has been highlighted in ***steel blue*** for clear visibility. We strongly recommend the reviewer to refer to these highlighted sections to better understand the adjustments and improvements we have made in response to the review comments during the rebuttal stage.
>
> We sincerely hope that the additional quantitative experiments, extended discussions, and updated manuscript collectively address your concerns. If you have any further questions or suggestions, please feel free to let us know — we would be more than happy to provide additional clarification or conduct further discussion.

---

> > ### Comment · Reviewer_pJC7 · 2025-11-24
> >
> > Thanks for the rebuttal and the requested experiments. I have adjusted the score.

---

> > > ### Author Response · Authors · 2025-11-25
> > >
> > > Thank you for reviewing our rebuttal and the additional experiments. Your constructive suggestions and feedback have genuinely helped us improve the quality of the paper. Any further guidance from you would also be sincerely appreciated!

---

### Author Response · Authors · 2025-11-29
**Rebuttal Summary for Area Chair**

Dear Area Chair,

We feel deeply sorry about the unexpected identity-leak accident that occurred during the rebuttal process. We would like to first clarify that we have never accessed, attempted to access, or been exposed to any reviewer identities in any form. We appreciate your additional time and effort in reassessing each submission under these circumstances, and we fully respect the work you are doing.

To support your evaluation, we have prepared this brief rebuttal summary outlining the main points discussed with the original reviewers. We structured our rebuttal in two stages: the first addressed questions that could be clarified immediately, and the second provided additional experimental results that required more time to complete. The current PDF reflects our revised version, where the ***steel blue*** highlighted parts indicate the refinements made during the rebuttal. For further details, please refer to our full discussions with the reviewers.

### Initial Responses
1. **Mask-related:** In response to [Reviewer pJC7-W2], [Reviewer 5Z3c-Q2], [Reviewer bJk4-Q1].

2. **Failure modes:** In response to [Reviewer 5Z3c-Q1], [Reviewer Wef7-Q1].

3. **Model generalization:** In response to [Reviewer Wef7-W1], [Reviewer bJk4-W2].

4. **Evaluation:** In response to [Reviewer 5Z3c-W2], [Reviewer Wef7-W2], [Reviewer bJk4-Q2].

### Experimental Updates
1. **Additional metrics:** In response to [Reviewer pJC7-W1], [Reviewer 5Z3c-W1], [Reviewer bJk4-W1].

2. **Ablation study between two phases:** In response to [Reviewer pJC7-Q1], [Reviewer Wef7-Q2], [Reviewer bJk4-W1].

3. **Ablation study on Extra Shape Augmentation:** In response to [Reviewer bJk4-Q1].

We would also like to note that 2 Reviewers—pJC7 and Wef7—**each increased their scores by 2 points** after we successfully addressed their concerns, resulting in updated scores of [6, 6, 8, 6]. All of these score improvements occurred before the identity-leak accident and were based solely on the technical discussions themselves, which can be verified from the reviewers’ official comments and the corresponding timestamps. We sincerely hope that this prior context can be taken into consideration as part of a fair and complete evaluation of our submission.

Once again, we thank the Area Chair for the effort, time, and care you have invested in reevaluating our work under such difficult circumstances. We truly appreciate your dedication to ensuring a fair and rigorous review process.

---

### Meta-Review · Area_Chair_HZor · 2026-01-05

**Summary:**

At a high level,  reviewers generally agreed that the paper is well-executed, clearly written, and demonstrates strong empirical performance on a challenging and practically relevant task, there was a recurring concern that the overall contribution is incremental rather than clearly transformative. Multiple reviewers felt that the core ideas—mask-guided swapping, adaptive masking, and multi-condition fusion—are reasonable and effective but largely represent careful engineering and data curation rather than a fundamentally new modeling paradigm. As a result, despite solid results and a newly introduced benchmark, several reviewers rated the paper as borderline, expressing uncertainty about whether the novelty and insight rise to a strong acceptance threshold.

More specifically, reviewers raised concerns about
(i) the evaluation method, particularly the reliance on VBench-style metrics that were shown to correlate weakly with human judgments in this task, motivating requests for additional semantic and identity-focused metrics;
(ii) insufficient quantitative ablations in the original submission, especially regarding the two-phase training scheme, adaptive masking details, and extra shape augmentation;
(iii) generalization limits, including sensitivity to mask quality, reliance on human-centric training data, and difficulty with cross-domain or out-of-distribution swapping

**Reviewer Concerns:**

In general,  the authors’ rebuttal and added experiments addressed many of these concerns. Hence, I plan to accept this paper

**Reviewer Scores:**

Three reviewer give accept (6) and One reviewer give reject (4)

---

### Decision · Program_Chairs · 2026-01-26

Accept (Poster)